# ALPHAZERO-LIKE TREE-SEARCH CAN GUIDE LARGE LANGUAGE MODEL DECODING AND TRAINING

## ABSTRACT

Large language models (LLMs) typically employ sampling or beam search, accompanied by prompts such as Chain-of-Thought (CoT), to boost reasoning and decoding ability. Recent work like Tree-of-Thought (ToT) and Reasoning via Planning (RAP) aim to augment the reasoning capabilities of LLMs by utilizing tree-search algorithms to guide multi-step reasoning. These methods mainly focus on LLMs' reasoning ability during inference and heavily rely on human-designed prompts to activate LLM as a value function, thus lacking general applicability and scalability. To address these limitations, we present an AlphaZero-like tree-search framework for LLMs (termed TS-LLM), systematically showing how tree-search with a learned value function can guide LLMs' decoding ability. TS-LLM distinguishes itself in two key ways: (1) Leveraging a learned LLM-based value function, our approach can be generally applied to different tasks beyond reasoning (such as RLHF alignment), and LLMs of any size, without prompting advanced, large-scale models. (2) It can guide LLM's decoding during both inference and training. Empirical evaluations across reasoning, planning, and RLHF alignment tasks validate the effectiveness of TS-LLM, even on trees with a depth of 64.

## 1 INTRODUCTION

Pretrained autoregressively on extensive corpora, large language models (LLMs) (OpenAI, 2023; Touvron et al., 2023a) have demonstrated their potential in a wide range of natural language tasks. A plethora of recent studies have concentrated on improving LLMs task-solving capability, including curation of larger and higher-quality general or domain-specific data (Touvron et al., 2023a; Zhou et al., 2023; Gunasekar et al., 2023; Feng et al., 2023; Taylor et al., 2022), more sophisticated prompt design (Wei et al., 2022; Zhou et al., 2022; Creswell et al., 2022), or better training algorithms with Supervised Learning or Reinforcement Learning (RL) (Dong et al., 2023; Gulcehre et al., 2023; Rafailov et al., 2023). When training LLMs with RL, LLMs' generation can be naturally formulated as a Markov Decision Process (MDP) and optimized with specific objectives. Following this formulation, ChatGPT (Ouyang et al., 2022) emerges as a notable success, optimizing LLMs to align human preference by leveraging RL from Human Feedback (RLHF) (Christiano et al., 2017).

Based on this formulation, LLMs' generation can be further guided with planning algorithms such as **tree search**. The success of AlphaZero (Silver et al., 2017b) has strongly proved the effectiveness of tree search algorithms, especially Monte Carlo Tree Search (MCTS)(Kocsis & Szepesvári, 2006; Coulom, 2006), in solving large-scale MDPs, including Atari and Go. It attracts recent efforts on the potential of leveraging these principled planning approaches to improve LLMs' reasoning ability.

Preliminary work in this field includes Tree-of-Thought (ToT) (Yao et al., 2023; Long, 2023) with depth/breadth-first search and Reasoning-via-Planing (RAP)(Hao et al., 2023) with MCTS. They successfully demonstrated a performance boost of searching on trees expanded by LLM through self-evaluation. Despite the strides, current methods come with distinct limitations. Firstly, these approaches mainly focus on enhancing LLMs' reasoning ability, lacking general applicability in different kinds of tasks, such as RLHF alignment. Secondly, as a model-based algorithm, the crucial reward or value function of tree-search algorithms is mainly obtained by prompting LLMs for step-level or trajectory-level reasoning. As a result, such algorithms lack their applicability and heavily rely on both well-designed prompts and large-scale LMs, such as GPT4 or LLaMA-33B. Only LLMs in such scale can provide reliably innate critic ability for reward and value estimation.

To address these problems, we introduce tree-search enhanced LLM (TS-LLM), an AlphaZero-like framework that utilizes tree-search to improve LLMs' performance on general natural language tasks. TS-LLM extends previous work to AlphaZero-like deep tree-search with a learned LLM-based value function and can guide LLM both during inference and training. Compared with previous work, TS-LLM has the following two new features:

- **TS-LLM offers a generally applicable and scalable pipeline.** It is generally **applicable**: (1) TS-LLM can be versatile to different tasks beyond reasoning. It can even solve RLHF alignment task in our experiment. (2) With a learned value function, TS-LLM can be applied to LLMs of any size. It does not require any well-designed prompts or advanced, large-scale LLMs. Our experiments show that TS-LLM can work for LLMs ranging from as small as 125M to as large as 7B. TS-LLM is also **scalable**: TS-LLM can conduct deep tree search, extending tree-search for LLM generation up to a depth of 64. This is far beyond 10 in ToT and 7 in RAP.

- **TS-LLM can potentially serve as a new LLM training paradigm beyond inference decoding.** By treating the tree-search operation as a policy improvement operator, we can conduct iterative processes of policy improvement and evaluation, to further train the LLM.

Furthermore, we present an in-depth analysis of the core design elements in TS-LLM, delving into the features, advantages, and limitations of different variations. We also offer a novel and reasonable evaluation metric to fairly compare tree-search algorithms with other baselines. Through comprehensive empirical evaluations on reasoning, planning, and RLHF alignment tasks, we benchmark the performance of various TS-LLM algorithm variants and underscore their superior capabilities. This showcases TS-LLM's potential as a universal framework to guide LLM decoding and training.

## 2 RELATED WORK

**Multistep Reasoning in LLMs** Multistep reasoning in language models has been widely studied, from improving the base model (Chung et al., 2022; Fu et al., 2023; Lewkowycz et al., 2022) to prompting LLMs step by step (Kojima et al., 2022; Wei et al., 2022; Wang et al., 2022; Zhou et al., 2022). Besides, a more relevant series of work has focused on enhancing reasoning through evaluations, including learned reward models (Uesato et al., 2022; Lightman et al., 2023) and self-evaluation (Shinn et al., 2023; Madaan et al., 2023). In this work, we apply evaluations to multistep reasoning tasks and token-level RLHF alignment tasks by learning a value function and reward model under the setting of a multistep decision-making process.

**Search guided reasoning in LLMs** While most CoT approaches have used a linear reasoning structure, recent efforts have been made to investigate non-linear reasoning structures such as trees (Jung et al., 2022; Zhu et al., 2023). More recently, various approaches for searching on trees have been applied to find better reasoning paths, e.g. beam search in Xie et al. (2023), depth-/breadth-first search in Yao et al. (2023) and Monte-Carlo Tree Search in (Hao et al., 2023). Compared with these methods, TS-LLM is a tree search guided LLM decoding and training framework with a learned value function, which is more generally applicable to both reasoning tasks and other scenarios like RLHF alignment tasks. Due to the limit of space, we leave a comprehensive comparison in Appendix A.

**Finetuning LLMs with Augmentation** Recent efforts have also been made to improve LLMs with augmented data. Rejection sampling is a simple and effective approach for finetuning data augmentation to improve LLMs' ability on single/multiple task(s) such as multistep reasoning (Yuan et al., 2023a; Zelikman et al., 2022) and alignment with human preference (Dong et al., 2023; Bai et al., 2022). Given an augmented dataset, reinforcement learning approaches have been also used to finetune LLMs (Gulcehre et al., 2023; Luo et al., 2023). Compared to previous works, TS-LLM leverages tree search as a policy improvement operator to generate augmented samples to train both the LLMs and the value function.

## 3 ENHANCING LARGE LANGUAGE MODEL WITH TREE SEARCH

In this section, we propose a versatile tree-search framework TS-LLM for guiding LLMs decoding and training. TS-LLM is summarized in the right part of Fig.1, and we conduct a systematic and comprehensive discussion about its key components. We start with the problem formulation with different tree/node designs, illustrating how LLMs' generation can be framed into a tree-search

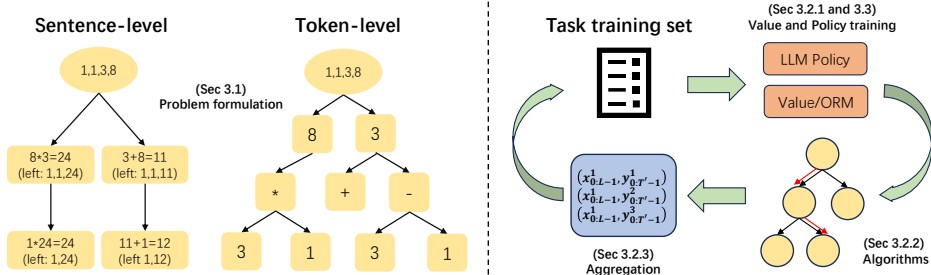

Figure 1: Left: Sentence/Token-level node on Game24 [1]. Right: Overall framework: TS-LLM is a generally applicable and scalable pipeline with multiple choices of tree search approaches and search aggregation methods. Firstly, with LLM finetuned on the task training set, TS-LLM generates trajectories to learn value/ORM. By treating tree search as a policy improvement operator, it iteratively augments the training set and further improves the LLM and value/ORM.

pipeline. Secondly, we show how tree-search can help with LLM inference, including choices of specific tree-search algorithms, value-function training techniques, and search aggregation methods. Building upon tree-search enhanced inference, we present how tree-search can serve as a general policy improvement operator for guiding LLMs training. Lastly, we discuss the tree-search algorithm's computation burdens for a fair comparison.

## 3.1 PROBLEM FORMULATION

We formulate the language generation process as a multi-step Markov Decision Process (MDP). The particularity of natural language tasks is that both the action and state are in language space. LLMs can serve as a policy $\pi_\theta$ that samples sequences of tokens as actions. Assuming the length of the output sequence and input prompt are $T$ and $L$ respectively, the probability for an LLM policy to produce an output sequence $\mathbf{y} = (y_0, y_1, \ldots, y_{T-1})$ conditioned on a prompt (input prefix) $\mathbf{x} = (x_0, x_1, \ldots, x_{L-1})$ is: $\pi_\theta(\mathbf{y}|\mathbf{x}) = \prod_{t=0}^{T-1} \pi_\theta(y_t|\mathbf{x}_{0:L-1}, \mathbf{y}_{0:t-1})$.

For a given natural language task, we can define a reward function $R(y_t|\mathbf{x}_{0:L-1}, \mathbf{y}_{0:t-1})$ as the task performance feedback for intermediate generation $y_t$ at timestep $t$. Due to the lack of large-scale and high-quality intermediate reward labels for general tasks, it is usually a sparse reward setting where any intermediate reward from the first $T - 1$ timestep is zero except the last $T$-th step. A typical case can be RLHF alignment task, where LLM can only receive the reward signal after it completes the full generation. Following the same logic, $\mathbf{y}$ can also be viewed as a sequence of sentences.

Given the problem formulation above, we successfully transfer the problem of better generation to optimization for higher cumulative reward. In this paper, we focus on how we can optimize it with **tree-search algorithms**. A specific natural language task typically predefines the state space (with language) and reward function (with task objective/metrics). What remains is the definition of action space, or in the context of tree-search algorithm, the action node design problem.

Tree search algorithms have validated their effectiveness for different action spaces in traditional RL research, including discrete action space (Silver et al., 2017a; Schrittwieser et al., 2020a) and continuous action space (Hubert et al., 2021). For conducting tree-search on LLMs, we categorize the action space design into the following two aspects:

- **Sentence-level action nodes**: For the tasks that have a step/sentence-level structure(e.g. chain-of-thought reasoning), it is natural to treat each thought as a sentence-level action node. Fig. 1 shows an example of sentence-level action nodes. This is also the technique adopted by ToT (Yao et al., 2023) and RAP (Hao et al., 2023). For each non-terminal node, the search tree is expanded by sampling serveral possible subsequent intermediate steps and dropping the duplicated generations.

- **Token-level action nodes**: Analogous to tree-search in discrete action space MDP, we can treat each token as a discrete action for LLM policy and the tree search can be conducted in token-level. We demonstrate an example of a Token-level action node in Fig. 1. For those tasks in which intermediate steps are not explicitly defined(e.g. RLHF alignment), splitting an output sequence into tokens might be a good choice.

---

[1]In our experiment on Game24, we only adopt the sentence-level node setting.

Note that for a proper state space, the node in both settings will be fixed once it is expanded.

Typically, the search space is determined by two algorithm-agnostic parameters, the **tree max width** $w$ and **tree max depth** $d$. In traditional RL, tree max-width typically refers to the size of action space while tree max-depth represents the episode length. In LLM generation, both action space designs have their own advantages and limitations over the search space. By splitting the generation into sentences, **sentence-level action nodes** provide a relatively shallow tree (low tree max-depth), simplifying the tree-search process. However, the large sample space of sentence-level generation makes full enumeration of all possible sentences infeasible. We have to set a maximum tree width $w$ to subsample $w$ nodes during the expansion, similar to the idea of Sampled MuZero(Hubert et al., 2021). Such subsampling results in the gap, determined by $w$, between the tree-search space and the LLM generation space. In addition, sentence-level action node often brings extra computation burdens, which will be further discussed in Section 3.4. For **token-level action nodes**, though it can get rid of the search space discrepancy and extra computational burdens, it greatly increases the depth of the tree, making tree-search more challenging.

## 3.2 Guiding LLM inference decoding with Tree Search

One of the benefits of tree-search algorithms is that they can optimize the cumulative reward by mere search, without any gradient calculation or update. Tree-search algorithms can enhance LLM generation without any further finetuning. In this section, given a fixed LLM policy, we present the full pipeline to illustrate how to guide LLM inference with tree search approaches.

### 3.2.1 Learning an LLM-based Value Function

As a model-based RL algorithm, tree search typically requires three models, including dynamic transition model $g$, reward model $\hat{r}$(Schrittwieser et al., 2020a), and value function $v$ Silver et al. (2017a). The dynamic transition in the LLM generation, given state $s_t = (\mathbf{x}_{0:L-1}, \mathbf{y}_{0:t-1})$ and action $a_t \sim \pi_\theta(\cdot|s_t)$, is known as: $g(s_t, a_t) = (\mathbf{x}_{0:L-1}, \mathbf{y}_{0:t})$. The value function $v$ and reward model $\hat{r}$ are the main issues. ToT and RAP obtain these two models by prompting advanced LLMs, such as GPT-4 or LLaMA-33B. As we will show in our experiment, specifically in Table 4, small LLM cannot serve as a reliable value model to guide the tree search process.

To make the tree search algorithm generally applicable, our method leverages a learned LLM-based value function $v_\theta(s)$ conditioned on state $s$ and a learned final-step outcome reward model (ORM) $\hat{r}_\theta$ (Uesato et al., 2022) since most tasks can be formulated as sparse-reward problems. Note that in fact the design of value function can be quite flexible and the essential aspect is its ability to consistently assess the state. Since we mainly deal with language-based task, we utilize a shared value network and reward model whose structure is a decoder-only transformer with an MLP to output a scalar on each position of the input tokens. And typically, LLM value's decoder is adapted from original LLM policy $\pi_\theta$'s decoder, or alternatively, the LLM value $v_\theta$ and policy $\pi_\theta$ can have a shared decoder (Figure 1.b in (Silver et al., 2017a)). For a sentence-level expanded intermediate step $s_t$, we use the prediction scalar at the last token as its value prediction $v_\theta(s_t)$. The final reward can be obtained at the last token when feeding the full sentences $(\mathbf{x}_{0:L-1}, \mathbf{y}_{0:T-1})$ into the model.

Therefore, we use language model $\pi_\theta$ as the policy to sample generations using the task training dataset. With true label or a given reward function in training data, a set of sampled tuple $\mathcal{D}_{\text{train}} = \{(\mathbf{x}^j, \mathbf{y}^j, r^j)\}_j$ of size $|\mathcal{D}_{\text{train}}|$ can be obtained, where $\mathbf{x}^j$ is the input text, $\mathbf{y}^j = s_{0:T^j-1}^j$ is the output text of $T^j$ steps and $r^j = R(\mathbf{y}^j|\mathbf{x}^j)$ is the ground-truth reward. Similar to the critic training in most RL algorithms, we construct the value target $z_t^j$ by TD-$\lambda$ or MC estimate on each single step $t$. The value network is optimized by mean squared error:

$$L(\theta) = \mathbb{E}_\mathcal{D} \left[ \sum_{t=0}^{T^j-1} ||v_\theta(s_{0:t}^j|\mathbf{x}^j) - z_t^j||_2^2 \right]. \tag{1}$$

The ORM $\hat{r}_\theta(\mathbf{y}_{0:T-1}|\mathbf{x}_{0:L-1})$ is learned with the same objective. Training an accurate value function and ORM is quite crucial for the tree-search process as they provide the main guidance during the search. We will further illustrate how different training conditions influence the learned value function, ORM, and corresponding tree-search performance in our experiment section.

### 3.2.2 Tree Search Algorithms

Given a learned value function, in this section, we present five types of tree-search algorithms. Due to the space limitation, we leave the background, preliminaries and detailed comparison of these tree-search algorithms in Appendix C.1.

**MCTS with value function approximation** (named as MCTS-$\alpha$): This is the MCTS variants utilized in AlphaZero (Silver et al., 2017a). Starting from the initial state, we choose the node of state $s_t$ as the root node and do several times of search simulations consisting of *select, expand and evaluate* and *backup*, where the leaf node value evaluated by the learned value function will be backpropagated to all its ancestor nodes. After the search, we choose an action proportional to the root node's exponentiated visit count, i.e. $a \sim \frac{N(s_t,a)^{1/\tau}}{\sum_b N(s_t,b)^{1/\tau}}$, and move to the corresponding next state. The above iteration will be repeated until finished. MCTS-$\alpha$ has two main features. Firstly, MCTS-$\alpha$ cannot trace back to its previous states once it takes an action. So it cannot restart the search from the initial state unless multiple searches are conducted which will be discussed in Section 3.2.3. Secondly, MCTS-$\alpha$ utilizes a value function so it can conduct the backward operation during the intermediate steps, without the need to complete the full generation to obtain a Monte-Carlo estimate.

**MCTS**: This approach was adopted in RAP (Hao et al., 2023), which refers to classic MCTS(Kocsis & Szepesvári, 2006). Specifically, the process starts from the initial state node and then *select* until finding a leaf node, if this node is non-terminal, expand it and repeat the previous *select* and *expand* steps. If the process encounters a terminal node, its value will be *evaluated* and *backup* to the whole path. In contrast to MCTS-$\alpha$, it only back-propagates the value on the terminal nodes, relying on a Monte-Carlo estimate of value, and it always starts searching from the initial state node.

**MCTS-Rollout**: Combining the features from the two algorithms above, we propose a new variant MCTS-Rollout for tree search. Similar to MCTS, MCTS-Rollout always starts from the initial state node. It further does the search simulations analogous to MCTS-$\alpha$, and the *backup* process can happen in the intermediate step with value function. It repeats the operations above until the process finds $N$ complete answers or reaches the computation limit (e.g. maximum number of tokens.) MCTS-Rollout can be seen as an offline version of MCTS-$\alpha$ so they may have similar application scope. The only difference is that MCTS-Rollout can scale up the token consumption for better performance since it always reconducts the search from the beginning.

**Breadth-first and Depth-first Search with value function based tree-pruning(BFS-V/DFS-V)**: These two search algorithms were adopted in ToT (Yao et al., 2023). The core idea is to utilize the value function to prune the tree for efficient search, while such pruning happens in tree breadth and tree depth respectively. BFS-V can be regarded as a beam-search with cumulative reward as the objective. In BFS-V, when setting beam size as $k^2$, nodes with $k$-largest values are maintained and expanded in each step. DFS-V chooses an unvisited node with the highest value until it finds a complete path. To improve the search efficiency, DFS-V prunes trees by either dropping the nodes whose value is below a predefined threshold or dropping the nodes of lower values.

### 3.2.3 Multiple Search and Search Aggregation

Inspired by Wang et al. (2022) and Uesato et al. (2022) that LLM can improve its performance on reasoning tasks by sampling multiple times and aggregating the candidates, TS-LLM also has the potential to aggregate $N$ complete answers generated by multiple tree searches.

When conducting multiple tree searches, we usually adopt **Intra-tree Search** setting. Intra-tree search conducts multiple tree searches on exactly the same tree, thus the state space is exactly the same. Such a method is computationally efficient as the search tree can be reused multiple times. However, the diversity of multiple generations might decrease because the former tree search might influence the latter tree searches. Also, the search space is limited by the sentence-level actions which are previously expanded in the tree and are fixed across multiple tree searches.

We refer to Appendix D.5 for an alternative setting called Inter-tree Search where we allow resampling in the expansion process. Without further specification, all settings in our paper are under the intra-tree search setting. Our next step is to aggregate these search results to obtain the final answer. With a learned ORM, we consider the following three different aggregation methods:

---

[2]For BFS-V, we can get $k$ complete answers by setting the beam size as $k$. Thus, $k = N$ in BFS.

Table 1: Task setups. The node, tree max with and tree max depth are search space parameters. Refer to Appendix D.6 and D.10 for how max tree-width and tree-depth are determined.

| Task | Category | Train/test size | Search Space Hyperparameters | | |
|------|----------|-----------------|------|------|------|
| | | | Node | Tree Max width | Tree Max depth |
| GSM8k | Mathematical Reasoning | 7.5k / 1.3k | Sentence | 6 | 8 |
| Game24 | Mathematical Planning | 1.0k / 0.3k | Sentence | 20 | 4 |
| PrOntoQA | Logical Reasoning | 4.5k / 0.5k | Sentence | 6 | 15 |
| RLHF | Alignment | 30k / 3k | Token | 50 | 64 |

**Majority Vote.** Wang et al. (2022) aggregates the answer $f$ by using majority vote over multiple random generation: $f^* = \arg\max_f \sum_{\mathbf{y}^j} \mathbf{1}_{\text{final\_ans}(\mathbf{y}^j)=f}$, where $\mathbf{1}$ is the indicator function.

**ORM-Max.** Given an outcome reward model, the aggregation can choose the answer $f$ with maximum final reward, $f^* = \text{final\_ans}(\arg\max_{\mathbf{y}^j} \hat{r}_\theta(\mathbf{y}^j|\mathbf{x}^j))$.

**ORM-Vote.** Given an outcome reward model, the aggregation can choose the answer $f$ with the sum of reward, namely $f^* = \arg\max_f \sum_{\mathbf{y}^j;\text{final\_ans}(\mathbf{y}^j)=f} \hat{r}_\theta(\mathbf{y}^j|\mathbf{x}^j)$

### 3.3 ENHANCING LLM TRAINING WITH TREE SEARCH

By plugging in the tree search operation, LLM can decode better compared to its original decoding performance during inference, which in other words, is a policy improvement process. Based on this, a new training/finetuning paradigm can be proposed, consisting of the following iterative process:

**Policy improvement**: We firstly generate tree-search trajectories based on $\pi_{\theta_{\text{old}}}$, $v_{\theta_{\text{old}}}$, and $\hat{r}_{\theta_{\text{old}}}$. By imitating new trajectories, LLM policy can be further improved to $\pi_{\theta_{\text{new}}}$.

**Policy evaluation**: Train value function $v_{\theta_{\text{new}}}$ and ORM $\hat{r}_{\theta_{\text{new}}}$ over the new sampled trajectories.

Such iterative process belongs to generalized policy iteration (Sutton & Barto, 2018), which is also the procedure used in AlphaZero's training. In our case, the training process involves finetuning three networks on the tree-search augmented training dataset.: (1) Policy network $\pi_\theta$: Use cross-entropy loss with trajectories' tokens as target (2) Value network $v_\theta$: Mean squared error loss with trajectories' temporal difference (TD) or Monte-Carlo (MC) based value estimation as target, and (3) ORM $\hat{r}_\theta$: Mean squared error loss with trajectories' final reward as target.

### 3.4 TREE SEARCH'S EXTRA COMPUTATION BURDENS

Tree-search algorithms will inevitably bring in additional computation burdens, especially in the node expansion phase for calculating legal child nodes and their corresponding value. Prior methodologies, such as ToT and RAP, tend to benchmark their performance against baseline algorithms using an equivalent number of generation paths (named Path@$N$). This approach overlooks the additional computational demands of the tree-search process and makes the comparison unfair.

In the setting of sentence-level action nodes, such computational demands are substantial. This is primarily because expanding a tree requires the generation of multiple sentences from a single parent node. In contrast, a token-level action node can alleviate the computation demand for all child nodes. A single forward inference can directly yield a probability distribution for all tokens so its search space and computational overhead align closely with the random sampling strategy typically used in LLM generation. Despite this, additional exploration steps (especially in Alphazero-like MCTS search) still introduce further computation. To enable a more fair comparison between tree-search and other baselines, we also should monitor the number of tokens generated for node expansion in our experiments. This provides a meaningful and fair comparison of different algorithms' performance when operating under comparable token generation conditions.

## 4 EXPERIMENTS

In this section, we first introduce detailed experimental setups in Section 4.1. In Section 4.2, we comprehensively present features of TS-LLM by answering six questions. Refer to Appendix D and E for more settings, implementations, results, and visualizations. Our code is open-sourced at `https://github.com/iclr2024tsllm/ICLR2024_TS_LLM`.

## 4.1 EXPERIMENT SETUPS

**Task setups.** For a given MDP, the nature of the search space is primarily characterized by two dimensions: depth and width. To showcase the efficacy of tree-search algorithms across varied search spaces, we evaluate all algorithms on four tasks with different search widths and depths, including the mathematical reasoning task GSM8k (Cobbe et al., 2021), mathematical planning task Game24 (Yao et al., 2023), logical reasoning task PrOntoQA (Saparov & He, 2022), and RLHF alignment task using synthetic RLHF data (Dahoas). The specific task statistics and search space hyperparameters are listed in Table 1. These hyperparameters, especially max search width and search depth, are determined by our exploration experiments. They can effectively present the characteristics of a task and define its search space. Refer to Appendix D.1 for more details of our evaluation tasks.

**Benchmark algorithms.** In our experiment, we benchmark all tree-search algorithms introduced in section 3.2.2, including MCTS-$\alpha$, MCTS-Rollout, MCTS, BFS-V, and DFS-V. Note that BFS-V, DFS-V, and MCTS are TS-LLM's variants instead of ToT(Yao et al., 2023) or RAP(Hao et al., 2023) baselines. The main difference is that we adopt a learned value function rather than prompting LLM. In our Question 3 and Table 4, we will present the results in ToT and RAP settings. In addition, we want to clarify that our experiments are not designed to validate our newly adopted MCTS-$\alpha$ and MCTS-Rollout are the new state-of-the-art among all search algorithms. We will present in our experiments that each search algorithm has its own task scope. We compare TS-LLM's variants with direct decoding methods: CoT greedy decoding, and CoT with self-consistency (Wang et al., 2022) (denoted as CoT-SC). Considering the search space gap between direct decoding and tree decoding (especially the sentence-level action node discussed in the section 3.1), we include the CoT-SC-Tree baseline which conducts CoT-SC over the tree's sentence nodes.

**Tree-search details.** Trees are expanded with sentence-level action nodes in GSM8k, Game24, and PrOntoQA, specifically, the output sequence is split into steps by '\n'. For GSM8k and PrOntoQA problems, we expand each node with at most 6 actions by LLM inference on the current state until outputs '\n' or '<EOS>', while for Game24 problems, we expand each node with at most 20 actions. We use token-level action nodes for RLHF aligment task. As we mentioned in section 3.2.2, such setting will result in a really deep search tree to a depth of 64.

**Model and training details.** For the rollout policy used in tree-search, we use LLaMA2-7B (Touvron et al., 2023b) on three reasoning tasks, and GPT-2-small (125M) on the RLHF alignment task. All LLMs will be first supervise-finetuned (SFT) on the tasks' training set, enabling their zero-shot CoT ability. Thus, **CoT in our experiments refers to zero-shot greedy decoding**. For value and ORM training, the data are generated by sampling the SFT policy's rollouts on the training set. Our policy LLM and value LLM are two separate models but are adapted from the same base model.

## 4.2 RESULTS AND DISCUSSIONS

**Question 1: How does TS-LLM perform in different generation tasks regarding the metric of Path@1/Path@$N$? Do Path@1/Path@$N$ provide a reasonable metric? (Sec. 3.2.2 and 3.4)**

In Table 2, we first provide the comparison between TS-LLM variants with the CoT baseline regarding the Path@1 performance. Path@1 means each algorithm only generates one answer. In this setting, the first three methods—MCTS, BFS-V, and DFS-V degenerate into the greedy search over value function [4]. We also refer readers to the **first** row of Fig 3 for Path@$N$ result, where MCTS, BFS-V and DFS-V are different. TS-LLM variants generally outperform the CoT baseline under Path@1/@N metric, showing that the value function provides effective guidance during LLM's decoding process. However, notable exceptions are also seen in the RLHF task where MCTS, BFS-V, and DFS-V underperform under Path@1/@N metric. From our results, we want to highlight two things. Firstly, MCTS, BFS-V, and DFS-V generally perform pretty well in shallow search problems (8 for GSM8K and 4 for Game24) while MCTS-$\alpha$ and MCTS-Rollout are dominant in searching on deep trees (15 for ProntoQA and 64 for RLHF). Secondly, the backward operation is quite important in deep search since MCTS, BFS-V, and DFS-V do not incorporate this operation in the Path@1 result.

---

[3] Generally, the tree-search state space changes with different seeds for random sampling and algorithm-dependent expansion orders. In the intra-tree setting, the tree node will be fixed once expanded.

[4] In Path@1 setting, we set beam size as 1 for BFS-V, search time as 1 for DFS and MCTS. MCTS@1 does not include the backward operation because it only happens after finding one complete path.

Table 2: Path@1 results of all TS-LLM variants and the CoT baseline with #token. For CoT-SC variants, we present Path@$N$, and $N$ is determined by maintaining the same level of token computation as TS-LLM's variants. For the alignment task, SC$_{ORM}$ means the best score of CoT-SC candidates, and SC$_{MAJ}$ refers to the average. CoT-SC-Tree also degenerates to CoT-SC in this task. MCTS/BFS-V/DFS-V degenerate to greedy value search, so the only uncertainty happens at node expansion. Sentence-level tasks still have variations by sampling while token-level task is deterministic.[3]

| Method | Performance(%) / # Tokens | | | | | | Reward / # Forward | |
|---|---|---|---|---|---|---|---|---|
| | GSM8k | | Game24 | | PrOntoQA | | RLHF(token-level) | |
| CoT-greedy | $41.4 \pm 0.0$ | 98 | $12.7 \pm 0.0$ | 76 | $48.8 \pm 0.0$ | 92 | $0.318 \pm 0.0$ | 57.8 |
| MCTS-$\alpha$ | $51.9 \pm 0.6$ | 561 | $63.3 \pm 1.9$ | 412 | $\mathbf{99.4 \pm 0.2}$ | 190 | $\mathbf{2.221 \pm 0.0}$ | 186 |
| MCTS-Rollout | $47.8 \pm 0.8$ | 3.4k | $\mathbf{71.3 \pm 2.5}$ | 670 | $96.9 \pm 0.6$ | 210 | $1.925 \pm 0.0$ | 809 |
| MCTS | $52.2 \pm 1.7$ | 486 | $64.0 \pm 3.2$ | 371 | $94.2 \pm 1.6$ | 125 | $-1.295 \pm 0.0$ | 61.8 |
| BFS-V | $\mathbf{52.5 \pm 1.3}$ | 485 | $64.8 \pm 2.9$ | 369 | $94.4 \pm 0.3$ | 126 | $-1.295 \pm 0.0$ | 61.8 |
| DFS-V | $51.8 \pm 0.6$ | 486 | $66.3 \pm 1.9$ | 369 | $93.3 \pm 0.8$ | 126 | $-1.295 \pm 0.0$ | 61.8 |
| CoT-SC$_{MAJ}$ | $46.8 \pm 1.1$ | 500 | $14.6 \pm 1.8$ | 684 | $61.1 \pm 1.4$ | 273 | $-0.253 \pm 0.01$ | 580 |
| CoT-SC$_{ORM}$ | $52.3 \pm 0.8$ | 500 | $50.6 \pm 2.0$ | 684 | $83.2 \pm 1.3$ | 273 | $1.517 \pm 0.01$ | 580 |
| CoT-SC-Tree$_{MAJ}$ | $45.8 \pm 1.3$ | 508 | $10.8 \pm 1.6$ | 651 | $59.8 \pm 0.7$ | 186 | - | - |
| CoT-SC-Tree$_{ORM}$ | $51.8 \pm 1.2$ | 508 | $47.9 \pm 1.4$ | 651 | $83.6 \pm 0.4$ | 186 | - | - |

Table 3: Path@1 metric on Game24 with different node size.

| Method | Performance(%) / # tokens | | | | | |
|---|---|---|---|---|---|---|
| | expand by 6 | | expand by 20 | | expand by 50 | |
| MCTS-$\alpha$ | 41.6 | 243 | 63.3 | 413 | 74.5 | 573 |
| Rollout | 43.8 | 401 | 71.3 | 670 | 80.7 | 833 |
| BFS-V | 43.2 | 206 | 64.8 | 370 | 74.6 | 528 |
| CoT-SC-Tree@10 | 38.8 | 508 | 48.3 | 656 | 48.3 | 707 |
| CoT-SC@10 | - | - | - | - | 52.9 | 0.8k |

Table 4: Results of TS-LLM variants on Game24 by prompting LLaMA2-7B as the value function (Setting used in ToT/RAP).

| Method | Performance(%) |
|---|---|
| CoT-greedy | $12.71 \pm 0.0$ |
| BFS@1 | $9.2 \pm 3.3$ |
| MCTS-$\alpha$@1 | $8.08 \pm 0.3$ |

Despite the superiority of TS-LLM, we argue that the metric Path@1/Path@$N$ may not be that reasonable. In table 2, we also include the number of computations used in Path@1 generation (number of tokens in sentence-level and number of forward computation in token-level). We also refer readers to the **second** row of Fig 3 for Path@$N$ result, with token/forward number as the x-axis. For Path@1/N, TS-LLM variants consume much more computation than the CoT baseline. To enable a fair comparison, we provide additional baselines, CoT-SC and CoT-SC-Tree with two aggregation methods: majority-vote (MAJ) and ORM-vote (denoted as ORM, and it utilizes the learned ORM in TS-LLM). In Table 2, we show results within a similar scale of computation consumption with TS-LLM variants. Under this situation, TS-LLM's advantages largely decrease when compared with CoT-SC$_{ORM}$, especially on GSM8K (only BFS greedy value search is the best). We are surprised to see that such simple algorithms can also have outstanding performance when compared fairly. Despite this, in Table 2 and the **second** row of Fig 3, most tree-search algorithms are still dominant in the rest three tasks given the same (CoT-SC-Tree) or larger search space (CoT-SC).

**Question 2: How do node expansion sizes and tree width influence the performance? (Sec. 3.1)**

As discussed in Sec. 3.1), the search space is limited by maximum tree width. Thus, we investigate the possible influence introduced by different tree constructions on Game24 with different node expansion sizes. Specifically, we set the number of maximal expanded node size as 6, 20, and 50. Table 3 lists the Path@1 performance and the number of tokens generated comparing TS-LLM's variants, CoT-SC and CoT-SC-Tree. The almost doubled performance boost from 43.8 to 80.7 indicates the impact of different expansion sizes and tree-width, improving TS-LLM's performance upper bound. In addition, we find BFS-V can show great performance with less token consumption under small tree-max-width setting (width=6), while MCTS-Rollout can finally be stronger by consuming more tokens (width=20/50). It inspires us that different tree-max-width and search space may also lead to different algorihtm superiority. The comparison among search algorithms should be conducted and concluded under the same search space. We refer the readers to Appendix D.6 for additional results on GSM8K and ProntoQA, where we further validate our conclusions in Q1/Q2.

**Question 3: Why do we need a learned value function and how to train that? (Sec. 3.2.1)**

Figure 2: Different Value training on GSM8k

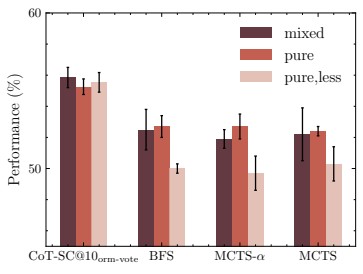

Table 5: Different value training for iterative update on GSM8k

| Method | Policy | Value | Performance(%) |
|---|---|---|---|
| MCTS-$\alpha$ | $\pi_{\theta_0}$ | $\{v, \hat{r}\}_{\theta_0}$ | $51.9 \pm 0.6$ |
| MCTS-$\alpha$ | $\pi_{\theta_0}$ | $\{v, \hat{r}\}_{\theta_1}^{\text{RL}}$ | $52.0 \pm 0.5$ |
| MCTS-$\alpha$ | $\pi_{\theta_0}$ | $\{v, \hat{r}\}_{\theta_1}$ | $53.2 \pm 0.3$ |
| MCTS-$\alpha$ | $\pi_{\theta_1}$ | $\{v, \hat{r}\}_{\theta_0}$ | $54.1 \pm 0.9$ |
| MCTS-$\alpha$ | $\pi_{\theta_1}$ | $\{v, \hat{r}\}_{\theta_1}^{\text{RL}}$ | $55.2 \pm 1.2$ |
| MCTS-$\alpha$ | $\pi_{\theta_1}$ | $\{v, \hat{r}\}_{\theta_1}$ | $\mathbf{56.5 \pm 0.6}$ |

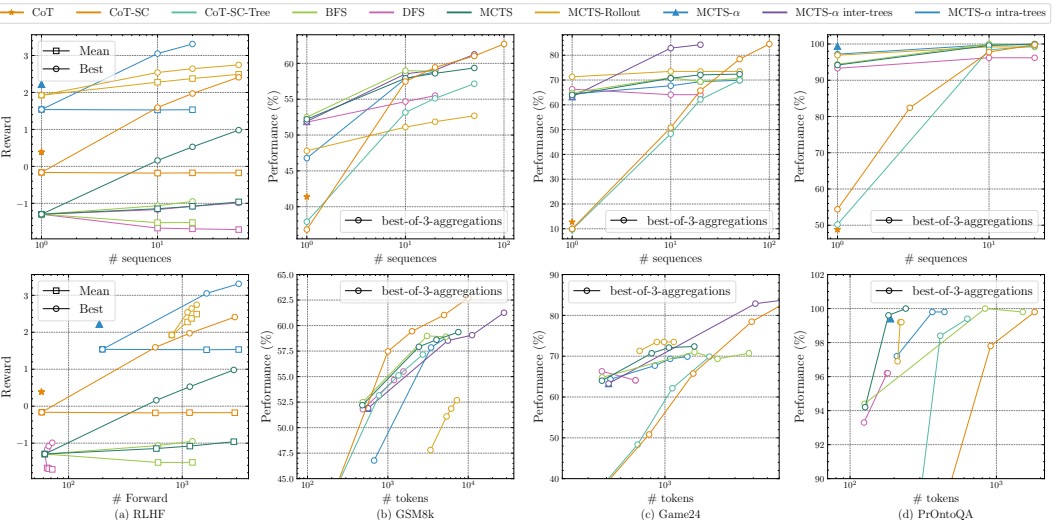

Figure 3: Mean/max reward for RLHF alignment and the best results of 3 aggregations for the rest tasks w.r.t. number of sequences (Path@$N$) on the 1st row and the number of tokens on the 2nd row.

In Table 4, we provide a motivating example by prompting LLaMA2-7b-base as the value function and reward model for TS-LLM on Game24. This is the setting used in ToT and RAP. The performance drop of BFS-V and MCTS-$\alpha$ indicates the incapability of small language models as reliable evaluators, which motivates us to substitute it with a learned value function and ORM as a general solution for any task and LLM with any size.

Therefore, we investigate data collection and training paradigms for value function and ORM in TS-LLM. In Figure 2, we investigate the influence of data amount and diversity by training with mixed data uniformly sampled from checkpoints of all SFT epochs (*mixed*); data purely sampled from the last checkpoint (*pure*); 1/3 data of the *pure* setting (*pure,less*). The results of CoT-SC$_{\text{ORM-vote}}$@10 underscore the diversity of sampled data in learning a better ORM. The Path@1 results of 3 TS-LLM variants show that the amount of sampled data is of great importance. We leave a detailed discussion of how value and reward function training is influenced in iterative training (Sec. 3.3) when answering Question 5. Our final conclusion is that **collecting a diverse dataset is better for the ORM and collecting as much data as possible is better for value function training.**

**Question 4: How TS-LLM is improved by aggregating over multiple results? (Sec. 3.2.3)**

In Fig. 3, we demonstrate the mean/max reward for the RLHF task and the best of 3 aggregation results for the rest three tasks. We measure the performance of aggregation w.r.t path number and token consumption. From the figure, we mainly summarize two conclusions: Firstly, **Most TS-LLM variants benefit from aggregation and can show large strengths compared with other baselines.** CoT-SC only beats TS-LLM in GSM8k with the same token size, mainly because of its larger search space. TS-LLM variants are still dominant when compared with CoT-SC-Tree. Secondly, **tree-search algorithms' aggregation benefits less than CoT-SC in small-scale problems.** In GSM8K and Game24, TS-LLM struggles to improve under large aggregation numbers. We believe this is because of: (1) The search space gap between CoT-SC and tree-search algorithms. Tree-search algorithms inherently explore fewer sentences, which is validated by comparing token consumption

Table 6: Iterative update results. $\theta_0$ is the old parameter while $\theta_1$ is the new one after one iteration. We compare all combinations of policy and value on GSM8k (left) and RLHF alignment (right).

| Method | Policy | Value | Performance(%) | Method | Policy | Value | Performance(%) |
|---|---|---|---|---|---|---|---|
| Greedy | $\pi_{\theta_0}$ | - | $41.4 \pm 0.0$ | Greedy | $\pi_{\theta_0}$ | - | $0.39 \pm 0.0$ |
| Greedy | $\pi_{\theta_1}$ | - | $\mathbf{47.9 \pm 0.0}$ | Greedy | $\pi_{\theta_1}$ | - | $1.87 \pm 0.0$ |
| Greedy | RFT k=50 | - | $47.0 \pm 0.0$ | Greedy | RFT N=5 | - | $1.16 \pm 0.0$ |
| Greedy | RFT k=100 | - | $47.5 \pm 0.0$ | Greedy | PPO | - | $\mathbf{2.53 \pm 0.0}$ |
| MCTS-$\alpha$ | $\pi_{\theta_0}$ | $\{v, \hat{r}\}_{\theta_0}$ | $51.9 \pm 0.6$ | MCTS-$\alpha$ | $\pi_{\theta_0}$ | $\{v, \hat{r}\}_{\theta_0}$ | $2.221 \pm 0.0$ |
| MCTS-$\alpha$ | $\pi_{\theta_0}$ | $\{v, \hat{r}\}_{\theta_1}$ | $53.2 \pm 0.3$ | MCTS-$\alpha$ | $\pi_{\theta_0}$ | $\{v, \hat{r}\}_{\theta_1}$ | $2.482 \pm 0.0$ |
| MCTS-$\alpha$ | $\pi_{\theta_1}$ | $\{v, \hat{r}\}_{\theta_0}$ | $54.1 \pm 0.9$ | MCTS-$\alpha$ | $\pi_{\theta_1}$ | $\{v, \hat{r}\}_{\theta_0}$ | $2.532 \pm 0.0$ |
| MCTS-$\alpha$ | $\pi_{\theta_1}$ | $\{v, \hat{r}\}_{\theta_1}$ | $\mathbf{56.5 \pm 0.6}$ | MCTS-$\alpha$ | $\pi_{\theta_1}$ | $\{v, \hat{r}\}_{\theta_1}$ | $\mathbf{2.670 \pm 0.0}$ |

between CoT-SC-Tree@50 and CoT-SC@50. (2) Different tree searches are not independent. The latter search might be influenced by the previous one, which decreases generation diversity.

**Question 5: Can TS-LLM further train LLM iteratively? (Sec. 3.3)**

To answer this question, we conduct initial experiments for one iterative update. We utilize MCTS-$\alpha$ with old policy $\pi_{\theta_0}$, value $v_{\theta_0}$ and ORM $\hat{r}_{\theta_0}$, to sample answers on the training dataset as an augmentation to the origin one. It will be further used to finetune these models to $(\pi_{\theta_1}, v_{\theta_1}, \hat{r}_{\theta_1})$. We include two baselines, RFT (Yuan et al., 2023b), which utilizes rejection sampling to finetune the policy, with different sampling numbers $k$ or top $N$, and PPO (Schulman et al., 2017) for the RLHF task. Note that PPO conducts multiple iterative updates. It is not fully comparable to our method and we only add it for completeness. Refer to Appendix D.8 and D.9 for more experimental details.

In Table. 6, we list results of iterative update on the GSM8K and RLHF, covering greedy decoding and MCTS-$\alpha$ over all policy and value combinations. Our empirical results validate that TS-LLM can further train LLM policy, value and ORM, boosting performance with the new policy $\pi_{\theta_1}$, new value and ORM $\{v, \hat{r}\}_{\theta_1}$, or both $(\pi_{\theta_1}, \{v, \hat{r}\}_{\theta_1})$ in CoT greedy decoding and MCTS-$\alpha$. $\pi_{\theta_1}$'s greedy performance is even slightly better than RFT which is specifically designed for GSM8k. We believe by further extending TS-LLM to multi-update, we can make it more competitive though currently $\pi_{\theta_1}$ still cannot beat PPO-based policy. For the training of value function and ORM (Question 3), we compare MCTS-$\alpha$ in Table 5. We train value and ORM in two paradigms, one ($\{v, \hat{r}\}_{\theta_1}$) is optimized from the initial weights and mixture of old and new tree-search data; another($\{v, \hat{r}\}_{\theta_1}^{\text{RL}}$) is optimized from $\{v, \hat{r}\}_{\theta_0}$ with only new tree-search data. This is called RL because training critic model in RL utilizes a similar process of continual training. The results show that $\{v, \hat{r}\}_{\theta_1}$ outperforms $\{v, \hat{r}\}_{\theta_1}^{\text{RL}}$ on both old and new policy when conducting tree search, contrary to the normal situation in traditional iterative RL training.

## 5    CONCLUSION

In this work, we propose TS-LLM, an LLM inference and training framework guided by Alphazero-like tree search that is generally versatile for different tasks and scaled to token-level expanded tree spaces. Empirical results validate that TS-LLM can enhance LLMs decoding and serve as a new training paradigm. We leave further discussion of our limitation and future work in Appendix B.

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

## A   MORE RELATED WORK AND COMPARISONS

Here we discuss the differences between TS-LLM and relevant work mentioned in Sec 2 in detail.

Recent efforts have been made to investigate non-linear reasoning structures such as trees(Jung et al., 2022; Zhu et al., 2023; Xu, 2023; Xie et al., 2023; Yao et al., 2023; Hao et al., 2023). Approaches for searching on trees with LLM's self-evaluation have been applied to find better reasoning paths, e.g. beam search in Xie et al. (2023), depth-/breadth-first search in Yao et al. (2023) and Monte-Carlo Tree Search in (Hao et al., 2023). Compared with these methods, TS-LLM is a tree search guided LLM decoding framework with a learned value function, which is more generally applicable to reasoning tasks and other scenarios like RLHF alignment. TS-LLM includes comparisons between different tree search approaches, analysis of computation cost, and shows the possibility of improving both the language model and value function iteratively. The most relevant work, CoRe (Zhu et al., 2023), proposes to finetune both the reasoning step generator and learned verifier for solving math word problems using MCTS for reasoning decoding which is the most relevant work to ours. Compared with CoRe, in this work TS-LLM distinguishes itself by:

**1.** TS-LLM is generally applicable to a wide range of reasoning tasks and text generation tasks under general reward settings, from sentence-level trees to token-level trees. But CoRe is proposed for Math Word Problems and only assumes a binary verifier (reward model).

**2.** In this work, we conduct comprehensive comparisons between popular tree search approaches on reasoning, planning, and RLHF alignment tasks. We fairly compare linear decoding approaches like CoT and CoT-SC with tree search approaches w.r.t. computation efficiency.

**3.** TS-LLM demonstrates potentials to improve LLMs' performance of direct decoding as well as tree search guided decoding, while in CoRe the latter cannot be improved when combining the updated generator (language model policy) with the updated verifier (value function) together.

Other related topic:

**Search guided decoding in LLMs** Heuristic search and planning like beam search or MCTS have also been used in NLP tasks including machine translation (Leblond et al., 2021) and code generation (Zhang et al., 2023; Matulewicz, 2022). During our preparation for the appendix, we find a concurrent work (Liu et al., 2023) which is proposed to guide LLM's decoding by reusing the critic model during PPO optimization to improve language models in alignment tasks. Compared with this work, TS-LLM focuses on optimizing the policy and value model through tree search guided inference and demonstrates the potential of continuously improving the policy and value models. And TS-LLM is generally applicable to both alignment tasks and reasoning tasks by conducting search on token-level actions and sentence-level actions.

## B   LIMITATION AND FUTURE WORK

Currently, our method TS-LLM still cannot scale to really large-scale scenarios due to the extra computation burdens introduced by node expansion and value evaluation. Additional engineering work such as key value caching is required to accelerate the tree-search. In addition, we do not cover all feasible action-space designs for tree search and it is flexible to propose advanced algorithms to automatically construct a tree mixed with both sentence-level expansion and token-level expansion, etc. We leave such exploration for future work. For MCTS aggregation, the current method still struggles to improve under large aggregation numbers. some new algorithms that can encourage multi-search diversity might be needed. Currently, we are still actively working on scaling up our method both during inference and training (especially multi-iteration training).

## C   BACKGROUNDS AND DETAILS OF EACH TREE-SEARCH ALGORITHMS IN TS-LLM

### C.1   PRELIMINARIES OF MONTE CARLO TREE-SEARCH ALGORIHTMS

Once we build the tree, we can use various search algorithms to find a high-reward trace. However, it's not easy to balance between exploration and exploitation during the search process, especially when the tree is sufficiently deep. Therefore we adopt Monte Carlo Tree Search(MCTS) variants as choices for strategic and principled search. Instead of the four operations in traditional MCTS

(Kocsis & Szepesvári, 2006; Coulom, 2006), we refer to the search process in AlphaZero (Silver et al., 2017a) and introduce 3 basic operations of a standard search simulation in it as follows, when searching actions from current state node $s_0$:

***Select*** It begins at the root node of the search tree, of the current state, $s_0$, and finishes when reaching a leaf node $s_L$ at timestep $L$. At each of these $L$ timesteps(internal nodes), an action(child node) is selected according to $a_t = \arg\max_a (Q(s_t, a) + U(s_t, a))$ where $U(s_t, a)$ is calculated by a variant of PUCT algorithm (Rosin, 2011):

$$U(s, a) = c_{\text{puct}} \cdot \pi_\theta(s, a) \frac{\sqrt{\sum_b N(s, b)}}{1 + N(s, a)} \tag{2}$$

where $N(s, a)$ is the visit count of selecting action $a$ at node $s$, and $c_{\text{puct}} = \log((\sum_b N(s, b) + c_{\text{base}} + 1)/c_{\text{base}}) + c_{\text{init}}$ is controlled by visit count and two constants. This search control strategy initially prefers actions with high prior probability and low visit count, but asymptotically prefers actions with high action-value.

***Expand and evaluate*** After encountering a leaf node $s_L$ by *select*, if $s_L$ is not a terminal node, it will be expanded by the language model policy. The state of the leaf node is evaluated by the value network, noted as $v(s_L)$. If $s_L$ is a terminal node, if there is an oracle reward function $R$, then $v(s_L) = R(s_L)$, otherwise, in this paper, we use an ORM $\hat{r}$ as an approximation of it.

***Backup*** After *expand and evaluate* on a leaf node, backward the statistics through the path $s_L, s_{L-1}, \ldots, s_0$, for each node, increase the visit count by $N(s_t, a_t) = N(s_t, a_t) + 1$, and the total action-value are updated as $W(s_t, a_t) = W(s_t, a_t) + v(s_L)$, the mean action-value are updated as $Q(s_t, a_t) = W(s_t, a_t)/N(s_t, a_t)$.

### C.2 Comparison of tree-search algorithms in TS-LLM

In this paper, we introduce three variants of MCTS based on the above basic operations. Among the 3 variants, MCTS-$\alpha$ is closer to AlphaZero(Silver et al., 2017a) and MCTS is closer to traditional Monte-Carlo Tree Search(Kocsis & Szepesvári, 2006). While MCTS-Rollout is closer to best-first search or A*-like tree search.

The major difference between the first three MCTS variants and BFS-V/DFS-V adopted from the ToT paper(Yao et al., 2023) is that the first three MCTS variants will propagate (i.e. the *backup* operation) the value and visit history information through the search process. MCTS-$\alpha$ and MCTS-Rollout bacpropagate after the *expand* operation or visiting a terminal node. MCTS backpropagates the information only after visiting a terminal node. For DFS-V, the children of a non-leaf node is traversed in a non-decreasing order by value. For efficient exploration, we tried 2 heuristics to prune the subtrees, (1) drop the children nodes with low value by *prune_ratio*. (2) drop the children nodes lower than a *prune_value*.

## D Experiment Details

### D.1 Task setups

**GSM8k** GSM8k (Cobbe et al., 2021) is a commonly used numerical reasoning dataset, Given a context description and a question, it takes steps of mathematical reasoning and computation to arrive at a final answer. There are about 7.5k problems in the training dataset and 1.3k problems in the test dataset.

**Game24** We also test our methods on Game24(Yao et al., 2023) which has been proven to be hard even for state-of-the-art LLMs like GPT-4. Each problem in Game24 consists of 4 integers between 1 and 13. And LLMs are required to use each number exactly once by $(+ - \times \div)$ to get a result equal to 24 We follow Yao et al. (2023) by using a set of 1362 problems sorted from easy to hard according to human solving time. We split the first 1k problems as the training dataset and the last 362 hard problems as the test dataset. For each problem in the training dataset, we collect data for SFT by enumerating all possible correct answers.

**PrOntoQA** PrOntoQA (Saparov & He, 2022) is a typical logical reasoning task in which a language model is required to verify whether a hypothesis is true or false given a set of facts and logical rules.

There are 4k problems in the training dataset and 500 problems in the test dataset.

**RLHF** We choose a synthetic RLHF dataset Dahoas[5] serving as the query data. We split the dataset to 30000/3000 as training and test set respectively. For the reward model, we choose reward-model-deberta-v3-large-v2[6] from OpenAssistant, which is trained from several RLHF datasets.

## D.2 SFT AND VALUE TRAINING DETAILS

**SFT in GSM8k, Game24 and PrOntoQA**: For GSM8k, Game24 and PrOntoQA, we first train LLaMA2-7b on the training dataset The training is conducted on 8 NVIDIA A800 GPUs, using a cosine scheduler decaying from lr=2e-5 to 0.0 with a warmup ratio of 0.03, batch size 128 for 3 epochs. For GSM8k and Game24 we use the checkpoint at the last epoch as the direct policy in experiments, while for PrOntoQA we use the checkpoint at the 1st epoch since the others overfit.

**Value training in GSM8k, Game24 and PrOntoQA**: Then we train the value function on the data rollout by the SFT policy. In GSM8k and Game24, For each model checkpoints of 3 epochs during SFT, we first collect 100 outputs per problem in the training dataset, then duplicate the overlapped answers, labeled each answer with our training set outcome reward ocracle. For data sampled by ech model checkpoint, we subsample 17 answers per problem, which is in total at most 51 answers per problem after deduplication. In PrOntoQA, we only sample 50 answers per problem with the first epoch model checkpoint and then do deduplication.

The value functions are trained in the same setting as supervised finetuning. We set the reward to be 1 when the output answer is correct and -1 otherwise. Then we use MC with $\gamma = 1$ to compute the returns. We do model selection on a validation dataset sampled from the direct policy model. For GSM8k, we train the value function and ORM for one epoch, while for Game24 and PrOntoQA we train the value function and ORM for 3 epochs.

**SFT in RLHF alignment**: We utilize GPT2-open-instruct[7], a GPT2-Small model supervised-finetuned over several instruction-tuning dataset.

**Value training in RLHF alignment**: Based on the SFT model, we collect 50 rollouts by the SFT policy for each question in the training set and label their final reward with the reward model. Then we train the value function and ORM for 2 epochs.

Note that here we start training the value function and ORM from the data sampled by the SFT policy model through direct decoding just as an initialization of the value function and ORM. After that TS-LLM can optimize the policy model, value function, and ORM simultaneously by adding new data sampled from tree search into the training buffer.

## D.3 DETAILS OF VALUE DATASET ABLATION

Here we introduce the details of building *mixed*, *pure* and *pure,less* datasets on GSM8k for value training in Fig 2. For each model checkpoints of 3 epochs during SFT, we first collect 100 outputs per problem in GSM8k training dataset, and then duplicate the overlapped answers, labeled each answer with our training set outcome reward ocracle. we sample multiple output sequences with temperature=0.7, top p=1.0 and top k=100.

*mixed* dataset: For each deduplicated dataset sampled by models of 3 epochs, we subsample 17 answers per problem.

*pure* dataset: we subsample 50 answers per problem from deduplicated dataset sampled by the last epoch policy model.

*pure,less* dataset: we subsample 17 answers per problem from deduplicated dataset sampled by the last epoch policy model.

For the results in Table 5, the details of training $\{v, \hat{r}\}_{\theta_1}$ can be find in Sec D.9. We use MC with $\gamma = 1$ to compute the returns. Here we describe the details of training $\{v, \hat{r}\}_{\theta_1}^{\text{RL}}$, we use the collected 78.7k samples in Sec D.8 to optimize $\{v, \hat{r}\}_{\theta_0}$. The training uses a cosine scheduler decaying from lr=2e-5 to 0.0 with a warmup ratio of 0.03, batch size 128 for 3 epochs.

---

[5]https://huggingface.co/datasets/Dahoas/synthetic-instruct-gptj-pairwise

[6]https://huggingface.co/OpenAssistant/reward-model-deberta-v3-large-v2

[7]https://huggingface.co/vicgalle/gpt2-open-instruct-v1

## D.4 DETAILS OF APPLYING EACH TREE SEARCH APPROACH

We present the implementation details and hyperparameters of all tree search approaches here.

Firstly, we refer to Table 1 for basic settings of each task. We set temperature=1.0, top_p=1.0, top_k=100 when using LLM to generate tree actions. To compute logprobs when expand actions in RLHF alignment task trees, we also use a temperature of 1.0.

For MCTS variants including MCTS-$\alpha$, MCTS-Rollout and MCTS, we need to define the hyperparamter in PUCT algorithm:

$$c_{\text{puct}} = \log((\sum_b N(s,b) + c_{\text{base}} + 1)/c_{\text{base}}) + c_{\text{init}} \quad (3)$$

In this paper, we fixed $c_{\text{base}}$ with 19652 and set $c_{\text{init}} = 3$ for GSM8k, Game24 and RLHF alignment tasks, set $c_{\text{init}} = 0.3$ for PrOntoQA tasks. Specifically, in MCTS-$\alpha$, we set the number of simulations before making an action to 5 for GSM8k, Game24 and PrOntoQA, 10 for RLHF alignment. We deterministically sampled actions with the largest visit count. And in MCTS-Rollout, we set an computation upperbound as number of generated tokens or number of model forwards, which is 51200 in GSM8k and Game24, 1800 in PrOntoQA and 5000 in RLHF alignment.

For DFS-V, the children of a non-leaf node is traversed in a non-decreasing order by value. For efficient exploration, we tried 2 heuristics to prune the subtrees, (1) drop the children nodes with low value by *prune_ratio*. (2) drop the children nodes lower than a *prune_value*. We set *prune_ratio* to be 0.7 for GSM8k, Game24 and PrOntoQA, and 0.9 for RLHF alignment task.

All Path@1 results for each tree search approach is conducted with 3 seeds and show the mean and standard deviation. Note that for Path@1 results of tree search approaches, the randomness comes from the node expansion process where we use an LLM to sample candidate actions. While for CoT-SC results, the randomness comes from sampling during direct decoding.

## D.5 DETAILS OF AGGREGATION RESULTS

Another alternative setting for conducting multiple searches in **Inter-tree Search**. Inter-tree Search builds a new tree for each new search, which increases the diversity of the search space, with extra computation burdens proportional to the search times. Thus, the intra-setting will have a larger state space compared with intra-tree setting. Our experiment results shown in Fig 3 (comparing MCTS-$\alpha$ intra-tree and inter-tree settings) also verify the performance gain brought by the larger search space.

We also present the details of how we sample multiple answers with tree search approaches and aggregate them into a final answer.

For the results of CoT-SC-Tree on Table 2 and Table 3, they can be viewed as **intra-tree** searches. For the results in Figure 3, only *MCTS-$\alpha$ inter-trees* were conducted with **inter-tree** searches, other tree-search algorithms (MCTS, MCTS-Rollout, BFS-V, DFS-V) were all conducted with **intra-tree** searches

Note that for all tree search algorithms except BFS-V, multiple searches are conducted in a sequential manner, while for BFS-V which can actually be regarded as Beam-Search, the number of searches means the number of beam size.

When sampling multiple intra-tree answers with MCTS-$\alpha$, we use a stochastic sampling setting. To ensure MCTS-$\alpha$ to explore sufficently, when selecting action of the current node, before doing several times of simulation, we add Dirichlet noise into the language model's prior probability of the current root node $s_0$, i.e. $\pi'_\theta(s_0, a) = (1 - \epsilon)\pi_\theta(s_0, a) + \epsilon\eta$, where $\eta \sim \text{Dir}(0.3)$, and we set $\epsilon = 0.25$ for both tasks. Actions are sample based on visit count, i.e. $a \sim \frac{N(s_t,a)^{1/\tau}}{\sum_b N(s_t,b)^{1/\tau}}$, where we set $\tau = 1$. After returning with a complete path, we clear the node statistics ($Q(s_t, a_t)$ and $N(s_t, a_t)$) on the tree to eliminate the influence of previous searches, while the tree structure is maintained. This setting is denoted as clear-tree when presented in the table. **In summary, when we only measure path@1 performance, we adopt MCTS-$\alpha$ (no sampling). But when we measure the aggregation performance, we use MCTS-$\alpha$-intra tree or MCTS-$\alpha$-inter tree. In MCTS-$\alpha$-intra tree we will activate the clear-tree and stochastic sampling setting.**

When sampling multiple answers with other tree-search methods, we only utilize the intra-tree aggregation variant, without stochastic sampling and clear-tree setting. This is because only MCTS-$\alpha$ and MCTS-rollout can conduct the above sampling and clear-tree operation. And we temporarily only apply such setting on MCTS-$\alpha$.

In ORM-vote, since we train the ORM with the reward signal -1 and 1, given a list of $N$ answers to be aggregated, we first normalize its values $\{\hat{r}(y^j)\}_j$ with min-max normalization make them in $[0, 1]$.

### D.6 RESULTS OF DIFFERENT NODE EXPANSION ON TASKS

Table 7, Table 8 and Table 9 show the path@1 results of MCTS-$\alpha$, MCTS-Rollout, BFS-V under different numbers of *tree-max-width* $w$ on GSM8k, Game24 and ProntoQA. And we also show the results of CoT-SC-Tree$_{ORM}$@10 and CoT-SC$_{ORM}$@10, which are aggregated by ORM-vote. The results are conducted under 3 seeds and we show the average value and standard deviation.

Let us first clarify how we choose the specific tree max width. For *tree-max-width* $w$ in GSM8k, Game24, and ProntoQA, we first start with an initial value $w = 6$. Then by increasing it (10 in GSM8K, 20 in Game 24, and 10 in ProntoQA), we can see the trends in performance and computation consumption. In ProntoQA/GSM8K, the performance gain is quite limited while the performance gain is quite large in Game24. So at last, we in turn try smaller *tree-max-width* in GSM8K/ProntoQA (3) and also try even larger *tree-max-width* (50) in Game24. Our final choice of $w$ (in Table 1) is based on the trade-off between the performance and the computation consumption. Currently, our selection is mainly based on empirical trials and it might be inefficient to determine the appropriate *tree-max-width* $w$. We think this procedure can be more efficient and automatic by comparing it with the results of CoT-SC on multiple samples to balance the tradeoff between performance and computation consumption. Because CoT-SC examples can already provide us with information about the model generation variation and diversity. We can also leverage task-specific features, e.g. in Game24, the correctness of early steps is very important, so a large $w$ can help to select more correct paths from the first layers on the search trees.

For the analysis of the results in Table 7, Table 8 and Table 9. We can mainly draw two conclusions aligned with that in Q1/Q2 in the main paper.

First, the overall trend that larger search space represents better tree-search performance still holds. For most tree-search settings, larger *tree-max-width* $w$ and search space bring in performance gain. The only exception happens at MCTS-Rollout on GSM8k decreases when the *tree-max-width* $w = 10$, this is due to the limitation of computation(limitation on the number of generated tokens which is 51200 per problem) is not enough in wider trees which results in more null answers. Despite the gain, the number of generated tokens also increases as the tree-max-width $w$ becomes larger.

Secondly, the conclusions in the main paper about comparing different search algorithms still hold. BFS performs pretty well in shallow search problems like (GSM8K/Game24). Though we can still see MCTS-$\alpha$ and MCTS-Rollout improve by searching in large *tree-max-width* (such as Game24 expanded by 50), the performance gain is mainly attributed to the extra token consumption and is quite limited. For deeper search problems like ProntoQA (15) and RLHF (64), the performance gap is obvious and more clear among all expansion widths. This aligns with our conclusion in Q1's analysis.

Table 7: Path@1 metric on GSM8k with different node size.

| Method | Performance(%) / # tokens | | | | | |
|---|---|---|---|---|---|---|
| | expand by 3 | | expand by 6 | | expand by 10 | |
| MCTS-$\alpha$ | $49.2 \pm 0.04$ | 460 | $51.9 \pm 0.6$ | 561 | $51.7 \pm 0.5$ | 824 |
| MCTS-Rollout | $47.2 \pm 0.8$ | 856 | $47.8 \pm 0.8$ | 3.4k | $45.9 \pm 0.9$ | 7.1k |
| BFS-V | $49.1 \pm 0.8$ | 260 | $52.5 \pm 1.3$ | 485 | $52.2 \pm 0.9$ | 778 |
| CoT-SC-Tree$_{ORM}$@10 | $52.4 \pm 1.2$ | 604 | $54.6 \pm 0.7$ | 780 | $54.5 \pm 1.1$ | 857 |
| CoT-SC$_{ORM}$@10 | - | - | - | - | $56.4 \pm 0.6$ | 1.0k |

Table 8: Path@1 metric on Game24 with different node size.

| Method | Performance(%) / # tokens | | | | | |
|---|---|---|---|---|---|---|
| | expand by 6 | | expand by 20 | | expand by 50 | |
| MCTS-$\alpha$ | $41.6 \pm 0.8$ | 243 | $63.3 \pm 1.9$ | 412 | $74.5 \pm 0.7$ | 573 |
| MCTS-Rollout | $43.8 \pm 5.3$ | 401 | $71.3 \pm 2.5$ | 670 | $80.7 \pm 1.5$ | 833 |
| BFS-V | $43.2 \pm 2.0$ | 206 | $64.8 \pm 2.9$ | 370 | $74.6 \pm 0.5$ | 528 |
| CoT-SC-Tree$_{ORM}$@10 | $38.8 \pm 2.0$ | 508 | $48.3 \pm 3.0$ | 656 | $48.3 \pm 4.2$ | 707 |
| CoT-SC$_{ORM}$@10 | - | - | - | - | $52.9 \pm 2.1$ | 0.8k |

Table 9: Path@1 metric on ProntoQA with different node size.

| Method | Performance(%) / # tokens | | | | | |
|---|---|---|---|---|---|---|
| | expand by 3 | | expand by 6 | | expand by 10 | |
| MCTS-$\alpha$ | $94.1 \pm 0.1$ | 151 | $99.4 \pm 0.2$ | 190 | $99.8 \pm 0.2$ | 225 |
| MCTS-Rollout | $85.9 \pm 0.8$ | 151 | $96.9 \pm 0.6$ | 210 | $99.3 \pm 0.4$ | 264 |
| BFS-V | $83.7 \pm 1.0$ | 105 | $94.4 \pm 0.3$ | 126 | $97.6 \pm 0.3$ | 145 |
| CoT-SC-Tree$_{ORM}$@10 | $91.9 \pm 0.8$ | 290 | $98.2 \pm 0.4$ | 417 | $99.1 \pm 0.1$ | 494 |
| CoT-SC$_{ORM}$@10 | - | - | - | - | $98.0 \pm 0.7$ | 0.9k |

## D.7 RESULTS PER TASK PER AGGREGATION

We show detailed results of GSM8k on Table 10, results of Game24 on Table 11, Table 12 for PrOnToQA and Table 13 for RLHF alignment. Due to the limit of computation resources, we show the results under 1 seed except for the path@1 results.

## D.8 SAMPLING DETAILS OF ITERATIVE UPDATE

We verify the idea of iteratively enhancing language model policy and value function model on the GSM8k and RLHF datasets.

**Sampling in GSM8k**: When sampling from the 7.5k problems in the GSM8k training dataset, we sample 12 sequences per problem in one sentence-level expanded tree, after deduplication, this results in 78.7k distinct answers, and 73.2% are correct answers. The sample parameters are listed in Table 14.

**Sampling in RLHF alignment**: We collect 10 answers for each training set problem sampled by MCTS-$\alpha$. We list the specific hyperparameters in Table 15.

To collect data for the rejection sampling baseline, we first sample 10 sequences per problem and then use the top 5 sequences for supervised fine-tuning.

## D.9 TRAINING DETAILS OF ITERATIVE UPDATE

**Policy training in GSM8k**: We construct the dataset for supervised finetuning by combining data in the training dataset with 57.6k correct answers sampled in Sec D.8 which results in 64.1k distinct correct answers. And we train the new policy model $\pi_{\theta_1}$ from the starting base model LLaMA2-7b for 3 epochs, following Yuan et al. (2023a). The training setting is the same as described in Sec D.2.

**Value training in GSM8k**: We construct the dataset for value and ORM training by combining the data used to train $\{v, \hat{r}\}_{\theta_0}$ with 78.7k answers sampled by MCTS-$\alpha$ in Sec D.8. To fairly compare $\{v, \hat{r}\}_{\theta_1}$ with $\{v, \hat{r}\}_{\theta_0}$, we drop samples in the former dataset to keep at most $51 - 12 = 39$ answers per problem resulting in 359k distinct answers. And we train the new value function $\{v, \hat{r}\}_{\theta_1}$ from the value model with its initial weight(before being updated on any data) for 3 epochs. The training setting is the same as described in Sec D.2.

**Policy training in RLHF alignment**: For the MCTS-$\alpha$'s training, we subsample the top 5 answers from the full 10 candidates (mentioned in Appendix D.8) to serve as the SFT dataset. For the RFT

Table 10: Detailed Results in GSM8k

| Method | N | Majority-vote | ORM-vote | ORM-max | #Token |
|---|---|---|---|---|---|
| CoT | - | 41.4 | 41.4 | 41.4 | 0.1k |
| CoT-SC | 1 | 38.21 | 38.21 | 38.21 | 0.1k |
| CoT-SC | 10 | 51.93 | 57.47 | 53.83 | 1k |
| CoT-SC | 20 | 54.44 | 59.44 | 54.74 | 2k |
| CoT-SC | 50 | 56.79 | 61.03 | 54.44 | 5k |
| CoT-SC | 100 | 58.15 | 62.70 | 53.68 | 10k |
| CoT-SC-Tree | 1 | 37.91 | 37.91 | 37.91 | 0.1k |
| CoT-SC-Tree | 10 | 50.19 | 53.15 | 50.95 | 0.8k |
| CoT-SC-Tree | 20 | 52.69 | 55.12 | 52.84 | 1.3k |
| CoT-SC-Tree | 50 | 54.51 | 57.16 | 53.45 | 2.7k |
| BFS-V | 1 | 52.5 | 52.5 | 52.5 | 0.5k |
| BFS-V | 10 | 58.98 | 56.25 | 54.97 | 3.1k |
| BFS-V | 20 | 58.91 | 56.79 | 52.39 | 5.3k |
| BFS-V | 50 | 59.29 | 59.36 | 53.22 | 10.1k |
| DFS-V | 1 | 51.8 | 51.8 | 51.8 | 0.5k |
| DFS-V | 10 | 53.75 | 54.66 | 53.37 | 1.2k |
| DFS-V | 20 | 54.89 | 55.50 | 53.53 | 1.6k |
| MCTS-$\alpha$ (no sampling) | 1 | 51.9 | 51.9 | 51.9 | 0.5k |
| MCTS-$\alpha$-intra tree | 1 | 46.78 | 46.78 | 46.78 | 0.7k |
| MCTS-$\alpha$-intra tree | 10 | 57.85 | 56.86 | 54.36 | 3.4k |
| MCTS-$\alpha$-intra tree | 20 | 58.83 | 58.23 | 55.19 | 5.3k |
| MCTS-$\alpha$-inter trees | 1 | 51.9 | 51.9 | 51.9 | 0.5k |
| MCTS-$\alpha$-inter trees | 10 | 57.92 | 58.53 | 55.34 | 5.5k |
| MCTS-$\alpha$-inter trees | 20 | 58.83 | 59.06 | 54.97 | 11.1k |
| MCTS-$\alpha$-inter trees | 50 | 58.76 | 61.26 | 53.98 | 27.8k |
| MCTS | 1 | 52.2 | 52.2 | 52.2 | 0.5k |
| MCTS | 10 | 57.92 | 55.72 | 53.75 | 2.4k |
| MCTS | 20 | 58.61 | 56.79 | 54.74 | 4.0k |
| MCTS | 50 | 59.36 | 58.23 | 53.75 | 7.5k |
| MCTS-Rollout | 1 | 47.8 | 47.8 | 47.8 | 3.4k |
| MCTS-Rollout | 10 | 51.10 | 50.49 | 49.81, | 5.4k |
| MCTS-Rollout | 20 | 51.86 | 51.25 | 50.19 | 6.1k |
| MCTS-Rollout | 50 | 52.69 | 52.24 | 50.49 | 7.2k |

n=5 baseline, we subsample the top 5 answers from 50 direct decodings as the SFT dataset. For the training of the PPO algorithm, we adopt the implementation from trlx[8]. We sample 20 answers for each question in total, which maintains the same level of token consumption during the PPO rollouts as that of MCTS-$\alpha$.

**Value training in RLHF alignment**: We construct the value and ORM dataset by mixing data from SFT-policy direct decoding and from MCTS-$\alpha$. To make the comparison fair, the new value function's training utilizes the same amount of data as the old one by subsampling 40 answers (from 50 shown in Appendix D.2) from direct decoding data and all 10 answers (shown in Appendix D.8) generated by MCTS-$\alpha$. We train our value function with learning rate 2e-5 and cosine scheduler from the initial model (instead of continuing training from the old value function) for 2 epochs.

## D.10 HYPERPARAMETER SELECTION PROTOCOLS

Here we present the selection protocols of hyperparameters.

---

[8]https://github.com/CarperAI/trlx

Table 11: Detailed Results in Game24

| Method | N | Majority-vote | ORM-vote | ORM-max | #Token |
|---|---|---|---|---|---|
| CoT | - | 12.7 | 12.7 | 12.7 | 0.1k |
| CoT-SC | 1 | 9.94 | 9.94 | 9.94 | 0.1k |
| CoT-SC | 10 | 13.54 | 50.83 | 50.83 | 0.8k |
| CoT-SC | 20 | 14.36 | 65.75 | 65.47 | 1.6k |
| CoT-SC | 50 | 16.30 | 78.45 | 78.45 | 4.0k |
| CoT-SC | 100 | 18.23 | 84.25 | 84.53 | 7.9k |
| CoT-SC-Tree | 1 | 9.67 | 9.67 | 9.67 | 0.1k |
| CoT-SC-Tree | 10 | 11.33 | 48.34 | 48.34 | 0.7k |
| CoT-SC-Tree | 20 | 13.26 | 61.60 | 62.15 | 1.1k |
| CoT-SC-Tree | 50 | 16.57 | 69.61 | 69.89 | 2.0k |
| BFS-V | 1 | 64.8 | 64.8 | 64.8 | 0.4k |
| BFS-V | 10 | 47.79 | 70.72 | 70.99 | 1.6k |
| BFS-V | 20 | 27.62 | 69.34 | 69.34 | 2.3k |
| BFS-V | 50 | 7.18 | 70.17 | 70.72 | 3.7k |
| DFS-V | 1 | 66.3 | 66.3 | 66.3 | 0.4k |
| DFS-V | 10 | 50.83 | 62.98 | 64.09 | 0.4k |
| DFS-V | 20 | 50.83 | 62.98 | 64.09 | 0.6k |
| MCTS-$\alpha$ (no sampling) | 1 | 63.3 | 63.3 | 63.3 | 0.4k |
| MCTS-$\alpha$-intra tree | 1 | 64.36 | 64.36 | 64.36 | 0.4k |
| MCTS-$\alpha$-intra tree | 10 | 66.85 | 67.68 | 63.90 | 0.9k |
| MCTS-$\alpha$-intra tree | 20 | 67.13 | 69.34 | 68.78 | 1.1k |
| MCTS-$\alpha$-intra tree | 50 | 67.96 | 69.89 | 69.34 | 1.4k |
| MCTS-$\alpha$-inter trees | 1 | 63.3 | 63.3 | 63.3 | 0.4k |
| MCTS-$\alpha$-inter trees | 10 | 72.65 | 82.87 | 82.32 | 4.1k |
| MCTS-$\alpha$-inter trees | 20 | 72.93 | 84.25 | 83.15 | 8.3k |
| MCTS | 1 | 64.0 | 64.0 | 64.0 | 0.4k |
| MCTS | 10 | 70.44 | 70.72 | 70.17 | 0.8k |
| MCTS | 20 | 72.10 | 72.10 | 71.27 | 1.1k |
| MCTS | 50 | 72.38 | 72.38 | 71.55 | 1.6k |
| MCTS-Rollout | 1 | 71.3 | 71.3 | 71.3 | 0.7k |
| MCTS-Rollout | 10 | 73.48 | 73.20 | 72.65 | 0.9k |
| MCTS-Rollout | 20 | 73.48 | 73.48 | 72.38 | 1.0k |
| MCTS-Rollout | 50 | 73.48 | 73.48 | 72.38 | 1.1k |

**Tree search general hyperparameters**: the *tree-max-depth $d$* limits the search depth of tree and *tree-max-width $w$* controls the max number of child nodes during node expansion. For the *tree-max-width $w$*, we refer the reader to Appendix D.6 for more discussions. We choose *tree-max-depth $d$* according to the statistics of the distribution of the number of steps from the dataset sampled by the LLM policy on the training set. Specifically, we statistically analyzed the number distribution of sentences in the training set, and in our experiments, these sentences are split by '\n'. For GSM8K, we set the *tree-max-depth $d$* at around the 99th percentile of the entire number distribution, to cover most query input and drop the outliers. Game24 has a fixed search depth of 4. For ProntoQA, we set the *tree-max-depth $d$* at the upper bound of the entire number distribution. For RLHF, this is not a reasoning task with CoT steps, so the depth can be flexible. We set it as the default value. In most cases, the depth of *tree-max-depth $d$* will not be reached. Because the node expansion will be terminated when we detect our pre-defined stop words in the generation (such as 'The answer is' or the '<EOS>' token).

**Specific hyperparameters for Monte Carlo Tree Search variants**: Basically we adopted the default values from Schrittwieser et al. (2020b) and Silver et al. (2017a) for most of the hyperparam-

Table 12: Detailed Results in PrOntoQA

| Method | N | Majority-vote | ORM-vote | ORM-max | #Token |
|---|---|---|---|---|---|
| CoT | - | 48.8 | 48.8 | 48.8 | 92 |
| CoT-SC | 1 | 54.40 | 54.40 | 54.40 | 91.25 |
| CoT-SC | 3 | 63.60 | 82.40 | 82.40 | 273.75 |
| CoT-SC | 10 | 58.40 | 97.80 | 97.80 | 912.55 |
| CoT-SC | 20 | 57.00 | 99.80 | 99.80 | 1.8k |
| CoT-SC-Tree | 1 | 50.20 | 50.20 | 50.20 | 82.02 |
| CoT-SC-Tree | 10 | 62.40 | 98.40 | 98.40 | 413.58 |
| CoT-SC-Tree | 20 | 61.00 | 99.40 | 99.40 | 632.91 |
| BFS-V | 1 | 94.40 | 94.40 | 94.40 | 125.52 |
| BFS-V | 10 | 99.00 | 100.00 | 100.00 | 837.78 |
| BFS-V | 20 | 98.60 | 99.80 | 99.80 | 1.5k |
| DFS-V | 1 | 93.30 | 93.30 | 93.30 | 124.46 |
| DFS-V | 10 | 95.60 | 96.20 | 96.00 | 178.79 |
| DFS-V | 20 | 95.60 | 96.20 | 96.20 | 180.92 |
| MCTS-$\alpha$ (no sampling) | 1 | 99.40 | 99.40 | 99.40 | 183.66 |
| MCTS-$\alpha$-intra tree | 1 | 97.20 | 97.20 | 97.20 | 208.68 |
| MCTS-$\alpha$-intra tree | 10 | 99.80 | 99.80 | 99.80 | 364.96 |
| MCTS-$\alpha$-intra tree | 20 | 99.80 | 99.80 | 99.80 | 441.31 |
| MCTS-$\alpha$-inter trees | 1 | 99.40 | 99.40 | 99.40 | 183.66 |
| MCTS-$\alpha$-inter trees | 10 | 100.00 | 100.00 | 100.00 | 1.9k |
| MCTS-$\alpha$-inter trees | 20 | 100.00 | 100.00 | 100.00 | 3.8k |
| MCTS | 1 | 94.20 | 94.20 | 94.20 | 126.65 |
| MCTS | 10 | 99.60 | 99.60 | 99.60 | 182.88 |
| MCTS | 20 | 100.00 | 100.00 | 100.00 | 240.16 |
| MCTS-Rollout | 1 | 96.90 | 96.90 | 96.90 | 210.41 |
| MCTS-Rollout | 10 | 99.20 | 99.20 | 99.20 | 220.16 |
| MCTS-Rollout | 20 | 99.20 | 99.20 | 99.20 | 224.16 |

eters, such as $c_{base} = 19652$ in Equation 3, $\tau = 1.0$ for MCTS-alpha stochastic search. And for the Dirichlet noise of MCTS-$\alpha$ stochastic search as mentioned in Appendix D.5, we adopted the default value in Silver et al. (2017a) as 0.3, which is specified for chess. We do find that in MCTS-$\alpha$, MCTS-Rollout and MCTS, $c_{init}$ can affect the balance between exploration and exploitation, and we chose it by running several trials from two possible values: $\{0.3, 3.0\}$. Moreover, for MCTS-$\alpha$, the hyperparameter *num of simulation*, $n_{simulation}$ is chosen as 5 for shallow trees (tree max-depths less than of equal to 15 over GSM8k, Game24 and ProntoQA) and 10 in deep trees(a tree max-depth of 64 in RLHF), controlling the search complexity at each step.

**Specific hyperparameters for BFS-/DFS-V**: BFS-V does not have hyperparameters for single search. For DFS-V, the children of a non-leaf node is traversed in a non-decreasing order by its value. For the sake of efficient exploration, we tried 2 heuristics to prune the subtrees, (1) drop the children nodes with lower values by *prune_ratio*. (2) drop the children nodes lower than a *prune_value*. The latter is adopted from Yao et al. (2023). In our experiments, we tried possible *prune_value*s from $\{0.5, 0.0, -0.5\}$ or `None`, we found that setting a high *prune_value* like 0.5 or 0.0 may introduce significant performance drop, however, setting a higher *prune_value* may introduce very closer answers. Therefore, we finally use *prune_ratio* for efficient exploration during searching on the tree with DFS-V. We set *prune_ratio* to be 0.7 for GSM8k, Game24 and PrOntoQA (tree-max-widths of 6, 6, 20), and 0.9 for RLHF alignment task since its much wider(a tree-max-width of 50).

Table 13: Detailed Results in RLHF alignment

| Method | N | Mean | Best | #Forward |
|---|---|---|---|---|
| CoT | 1 | 0.387 | 0.387 | 57.8 |
| CoT-SC | 1 | -0.164 | -0.164 | 58 |
| CoT-SC | 10 | -0.182 | 1.592 | 0.6k |
| CoT-SC | 20 | -0.175 | 1.972 | 1.2k |
| CoT-SC | 50 | -0.176 | 2.411 | 2.9k |
| BFS-V | 1 | -1.295 | -1.295 | 61.8 |
| BFS-V | 10 | -1.523 | -1.065 | 0.6k |
| BFS-V | 20 | -1.520 | -0.948 | 1.2k |
| BFS-V | 50 | -1.474 | -0.813 | 3.1k |
| DFS-V | 1 | -1.295 | -1.295 | 61.8 |
| DFS-V | 10 | -1.67 | -1.17 | 64.8 |
| DFS-V | 20 | -1.69 | -1.08 | 66.9 |
| DFS-V | 50 | -1.71 | -0.99 | 71.9 |
| MCTS-$\alpha$ (no sampling) | 1 | 2.221 | 2.221 | 186 |
| MCTS-$\alpha$-intra tree | 1 | 1.538 | 1.538 | 198.50 |
| MCTS-$\alpha$-intra tree | 10 | 1.527 | 3.052 | 1.6k |
| MCTS-$\alpha$-intra tree | 20 | 1.533 | 3.311 | 3.1k |
| MCTS | 1 | -1.295 | -1.295 | 61.8 |
| MCTS | 10 | -1.146 | 0.160 | 0.6k |
| MCTS | 20 | -1.08 | 0.528 | 1.2k |
| MCTS | 50 | -0.961 | 0.981 | 2.8k |
| MCTS-Rollout | 1 | 1.925 | 1.925 | 0.8k |
| MCTS-Rollout | 10 | 2.278 | 2.540 | 1.1k |
| MCTS-Rollout | 20 | 2.376 | 2.643 | 1.2k |
| MCTS-Rollout | 50 | 2.491 | 2.746 | 1.3k |

Table 14: Hyperparameters of sampling in GSM8k for LLM decoding(left), tree construction setting(middle), and MCTS-$\alpha$ setting(right).

| Hyperparameter | value |
|---|---|
| temperature | 1.0 |
| top_p | 1.0 |
| top_k | 100 |

| Hyperparameter | value |
|---|---|
| Tree Max width | 6 |
| Tree Max depth | 8 |
| Node | Sentence |

| Hyperparameter | value |
|---|---|
| num simulation | 5 |
| clear tree | True |
| stochastic sampling | True |
| $c_{base}$ | 19652 |
| $c_{init}$ | 3 |
| $\tau$ | 1.0 |

Table 15: Hyperparameters of sampling in RLHF alignment for LLM decoding(left), tree construction setting(middle), and MCTS-$\alpha$ setting(right).

| Hyperparameter | value |
|---|---|
| temperature | 1.0 |
| top_p | 1.0 |
| top_k | 50 |

| Hyperparameter | value |
|---|---|
| Tree Max width | 50 |
| Tree Max depth | 64 |
| Node | Token |

| Hyperparameter | value |
|---|---|
| num simulation | 5 |
| clear tree | True |
| stochastic sampling | True |
| $c_{base}$ | 19652 |
| $c_{init}$ | 3 |
| $\tau$ | 1.0 |

## D.11 WALL-TIME AND ENGINEERING CHALLENGES

Table 16, Table 17 and Table 18 show the wall-time of running different tree search algorithms implemented in TS-LLM , searching for one answer per problem (i.e. path@1). We also show the

wall-time of CoT greedy decoding and CoT-SC@10 with ORM aggregation as comparisons. We record the wall-time of inferencing over the total test dataset of each task.

The experiments were conducted on the same machine with 8 NVIDIA A800 GPUs, the CPU information is Intel(R) Xeon(R) Platinum 8336C CPU @ 2.30GHz.

We can find that the comparisons of wall-time within all the implemented tree-search algorithms in TS-LLM are consistent with those of the number of generated tokens. However, compared to CoT greedy decoding, the wall-time results of most tree search algorithms in TS-LLM are between two and three times of CoT greedy decoding's wall-time, excepting MCTS-Rollout runs for a very long time on GSM8k. And when comparing the wall-time and number of generated tokens between the tree-search methods and CoT-SC$_{ORM}$@10, TS-LLM is not as computationally efficient as CoT-SC decoding due to the complicated search procedures and extra computation introduced by calling value functions in the intermediate states.

There are two more things we want to clarify. Firstly, as we mentioned in Appendix B, our current implementation only provides an algorithm prototype without specific engineering optimization. We find a lot of repeated computations are performed in our implementation when evaluating the child node's value. Overall, there still exists great potential to accelerate the tree-search process, which will be discussed in the next paragraph. We are continuously working on this (we will discuss the engineering challenges in the next paragraph). Secondly, this wall-time is just Path@1 result so they have different token consumptions, and DFS-V, BFS-V, and MCTS will degenerate into greedy value search as we mentioned before. We are also working on monitoring the time consumption for Path@N results so we can compare them when given the same scale of token consumption.

**Engineering challenges and potentials.** Here we present several engineering challenges and potentials to increase the tree-search efficiency.

- **Policy and Value LLM with the shared decoder.** Our current implementation utilizes separate policy and value decoders but a shared one might be a better choice for efficiency. If so, most extra computation brought by value evaluation can be reduced to simple MLP computation (from the additional value head) by reusing computation from LLM's policy rollout. It can largely increase the efficiency. We only need to care about the LLM's rollout computations under this setting.

- **KV cache and computation reuse.** KV cache is used in most LLM's inference processes such as the Huggingface transformer's (Wolf et al., 2020) generation function. It saves compute resources by caching and reusing previously calculated key-value pairs in self-attention. In the tree search problem, when expanding or evaluating a node, all preceding calculations for its ancestor nodes can be KV-cached and reused. However, because of the large state space of tree nodes, we cannot cache all node calculations since the GPU memory is limited and the communication between GPU/CPU is also inefficient (if we choose to store such cache in CPU). More engineering work is needed to handle the memory and time tradeoff.

- **Large-batch vectorization.** Currently, our node expansion and node evaluation are only vectorized and batched given one parent node. We may conduct batch inference over multiple parent nodes for large-batch vectorization when given enough computing resources.

- **Parallel tree-search over multi-GPUs.** Our implementation handles each tree over a single GPU. AlphaZero (Silver et al., 2017a) leverages parallel search over the tree(Segal, 2010), using multi-thread search to increase efficiency. In LLM generation setting, the main bottleneck comes from the LLM inference time on GPU. Thus more engineering work is needed for conducting parallel tree-search over multi-GPUs.

- **Tree-Search with speculative decoding** Speculative decoding (Leviathan et al., 2023) is a pivotal technique to accelerate LLM inference by employing a smaller draft model to predict the target model's outputs. During the speculative decoding, the small LLM gives a generation proposal while the large LLM is used to evaluate and determine whether to accept or reject the proposal. This is similar to the tree-search process with value function pruning the sub-tree. There exists potential that by leveraging small LLMs as the rollout policy while large LLMs as the value function, we can also have efficient tree-search implementations.

Table 16: Wall-time results on GSM8k

| Method | Overall Time(sec) | Average Time(sec) | #Average Token |
|---|---|---|---|
| CoT-Greedy | 216.93 | 0.17 | 98 |
| CoT-SC$_{ORM}$@10 | 479.03 | 0.37 | 1k |
| MCTS | 378.13 | 0.29 | 486 |
| MCTS-$\alpha$ | 527.31 | 0.41 | 561 |
| MCTS-Rollout | 2945.94 | 2.27 | 3.4k |
| BFS-V | 383.08 | 0.29 | 485 |
| DFS-V | 387.00 | 0.30 | 486 |

Table 17: Wall-time results on Game24

| Method | Overall Time(sec) | Average Time(sec) | #Average Token |
|---|---|---|---|
| CoT-Greedy | 44.88 | 0.15 | 76 |
| CoT-SC$_{ORM}$@10 | 86.53 | 0.29 | 0.8k |
| MCTS | 81.22 | 0.27 | 371 |
| MCTS-$\alpha$ | 134.19 | 0.45 | 412 |
| MCTS-Rollout | 193.18 | 0.64 | 670 |
| BFS-V | 79.48 | 0.26 | 369 |
| DFS-V | 80.46 | 0.27 | 369 |

Table 18: Wall-time results on ProntoQA

| Method | Overall Time(sec) | Average Time(sec) | #Average Token |
|---|---|---|---|
| CoT-Greedy | 74.09 | 0.15 | 77 |
| CoT-SC$_{ORM}$@10 | 218.12 | 0.44 | 0.8k |
| MCTS | 130.15 | 0.26 | 125 |
| MCTS-$\alpha$ | 236.26 | 0.47 | 190 |
| MCTS-Rollout | 238.62 | 0.48 | 210 |
| BFS-V | 130.35 | 0.26 | 126 |
| DFS-V | 130.47 | 0.26 | 126 |

## D.12 DISCUSSION ABOUT SHARED LLM DECODER FOR BOTH POLICY AND CRITIC.

As we mentioned in Appendix D.11, using a shared decoder for the policy and value LLM might further improve the computation efficiency for the tree-search process. Therefore, we conducted an ablation to compare the model under the settings of a *shared decoder* and the setting of *separated decoder*s on Game24.

We first describe the training setting of both types of models we compared. For the setting of *separated decoder*, we refer to Appendix D.2 for details about dataset and training hyperparameters. For the setting of *shared decoder*, we train the shared policy and value LLM with the same data used in the setting of *separated decoder*. During training, a batch from the supervised finetuning (SFT) dataset and a batch from the value training dataset are sampled, the total loss of the shared policy and value LLM is computed by $\mathcal{L}_{total} = \mathcal{L}_{SFT} + 0.5 \cdot \mathcal{L}_{Value}$, where $\mathcal{L}_{SFT}$ is the cross entropy loss of predicting the next token in the groundtruth answer and $\mathcal{L}_{Value}$ is the Mean Square Error loss as we described in Equation 1. The training is conducted on 8 NVIDIA A800 GPUs, using a cosine scheduler decaying from lr=2e-5 to 0.0 with a warmup ratio of 0.03, a batch size of 128 for the supervised finetuning dataset and a batch size of 128 for the value training dataset. And we trained the shared decoder model for 3 epochs. This training setting is the same as used in the *separated decoder* setting.

We will compare these two types of model in the perspective of performance and computation efficiency. All tree-search algorithms are conducted under the same hyperparameters as those in Table 2 on Game24 in which the tree-max-width $w$ is set to 20.

Table 19 shows the comparisons of the two types of model on performance. Though the performance of CoT (CoT greedy decoding) of *shared decoder* model increases from 12.7 to 16.3, the number of tokens generated per problem also increases greatly from 76 to 166. By checking the models' outputs, we find the *shared decoder* model doesn't always obey the rules of Game24(There are only 4 steps of calculations and each number must be used exactly once). It usually outputs multiple steps, more than the four steps required in Game24. This might be regarded as hallucination problem which happens more frequently than in the *separated decoder* model. For the results of CoT-SC-Tree$_{ORM}$@10 (search by LLM's prior on trees and aggregated by ORM-vote), we observe close results of CoT-SC$_{ORM}$@10 Meanwhile, for the results of MCTS-$\alpha$, MCTS-Rollut, BFS-V and CoT-SC-Tree$_{ORM}$@10, there is only a small difference between the performance of the two models. We also observe an increase in the number of token generated per problem. This is because the *shared decoder* model is prone to output more invalid answers than the *separated decoder* model. Therefore, there are more distinct actions proposed in the last layers of the trees.

Next, we show some preliminary comparisons in Table 20 on the computation efficiency of the *separated decoder* model and the *shared decoder* model, from the results of expanding a tree node and evaluating its children. Specifically, Table 20 presents the node expansion time and value calculation time with/without KV Cache under token-level and sentence-level situations. For the token-level node, we set $w = 50$ while for the sentence-level node, we set $w = 20$. The results successfully present that a shared decoder can largely increase the computational efficiency for value estimation (20x in the token-level setting and 9x in the sentence-level setting).

In all, in this section, we initially conduct explorations on leveraging shared policy and value LLM decoder. The result proves the potential of computational efficiency for the shared structure. However, more work is needed to help the stability of policy/value performance.

Table 19: Comparisons of separated/shared LLM decoder policy and critic models on Game24

| Method | Performance(%) / # tokens | | | |
|---|---|---|---|---|
| | Separated Decoder | | Shared Decoder | |
| CoT | 12.7 | 76 | 16.3 | 166 |
| CoT-SC$_{ORM}$@10 | $52.9 \pm 2.1$ | 0.8k | $52.8 \pm 2.4$ | 1.9k |
| MCTS-$\alpha$ | $63.3 \pm 1.9$ | 412 | $64.1 \pm 1.3$ | 561 |
| MCTS-Rollout | $71.3 \pm 2.5$ | 670 | $70.6 \pm 0.4$ | 855 |
| BFS-V | $64.8 \pm 2.9$ | 370 | $63.0 \pm 1.0$ | 495 |
| CoT-SC-Tree$_{ORM}$@10 | $48.3 \pm 3.0$ | 656 | $45.5 \pm 2.0$ | 745 |

Table 20: Time (seconds) on policy expansion and value evaluation for a single tree node. When using a shared LLM decoder for policy and value LLM, we can use KV Cache for value calculation. It is much more efficient than a separate value decoder without KV cache.

| Node Type | Policy Expansion | Value with Cache | Value without Cache |
|---|---|---|---|
| Token-Level | 0.067 | 0.074 | 2.02 |
| Sentence-Level | 0.165 | 0.122 | 1.03 |

# E QUALITATIVE RESULTS

Here we present some qualitative results of generation for different methods in Table 21 (for Game24), Table 22 (for GSM8k), Table 23 (for ProntoQA) and 24 (for the RLHF alignment task).

| Problem: 4 8 9 13 | | |
|---|---|---|
| Methods | Response | Correctness |
| CoT | 8 + 13 = 21 (left: 4 9 21)
4 - 9 = -5 (left: -5 21)
21 - -5 = 26 (left: 26)
The answer is (8 + 13) - (4 - 9) = 26 | No |
| BFS-V | 9 + 3 = 12 (left: 4 8 12)
12 / 4 = 3 (left: 3 8)
3 * 8 = 24 (left: 24)
The answer is (9 + 3) / 4 * 8 = 24 | No |
| MCTS-$\alpha$ | 13 - 9 = 4 (left: 4 4 8)
4 * 4 = 16 (left: 8 16)
16 + 8 = 24 (left: 24)
The answer is 4 * (13 - 9) + 8 = 24 | Yes |
| MCTS-Rollout | 13 - 9 = 4 (left: 4 4 8)
4 * 4 = 16 (left: 8 16)
16 + 8 = 24 (left: 24)
The answer is (13 - 9) * 4 + 8 = 24 | Yes |

Table 21: Qualitative examples of Game24 outputs

| Question: A new program had 60 downloads in the first month. The number of downloads in the second month was three times as many as the downloads in the first month, but then reduced by 30% in the third month. How many downloads did the program have total over the three months? | | |
|---|---|---|
| Methods | Response | Correct |
| CoT | The number of downloads in the second month was 3*60 = 180
The number of downloads in the third month was 180*0.30 = 54
The total number of downloads for the three months was 60+180+54 = 294
The answer is 294 | No |
| BFS-V | In the second month, the program had 60*3=180 downloads
In the third month, the program had 180*30/100=54 downloads which was 30% less than the previous month
Total downloads for the program in the three months was 60+180+54=294
The answer is 294 | No |
| MCTS-$\alpha$ | The program had 60 x 3 = 180 downloads in the second month.
The downloads in the third month was 180 x 0.30 = 54 downloads lesser than the previous month.
So, the program had 180 - 54 = 126 downloads in the third month.
Therefore, the program had 60 + 180 + 126 = 366 downloads over the three months.
The answer is 366 | Yes |
| MCTS-Rollout | The number of downloads in the second month was 3 times as many as the downloads in the first month, which means the program had 3*60 = 180 downloads in the second month.
The total number of downloads for the two months was 180+60 = 240 In the third month, the program had 180-30/100*180 = 120 downloads, 30% of the number of downloads it had in the second month.
The total number of downloads for the three months was 240+120 = 360
The answer is 360 | No |

Table 22: Qualitative examples of GSM8k outputs

| Question: Butterflies are lepidopterans. Every arthropod is small. Whales are not small. Invertebrates are animals. Every insect is an arthropod. Lepidopterans are insects. Every insect is six-legged. Every arthropod is an invertebrate. Animals are multicellular. Polly is a lepidopteran. Is the statement "Polly is not small" true or false? | | |
| --- | --- | --- |
| Methods | Response | Correct |
| CoT | Polly is a lepidopteran. Lepidopterans are insects. Polly is an insect. Every insect is an arthropod. Polly is an arthropod. Every arthropod is small. Polly is small. The answer is true. | No |
| BFS-V | Polly is a lepidopteran. Lepidopterans are insects. Polly is an insect. Every insect is an arthropod. Polly is an arthropod. Every arthropod is small. Polly is small. Polly is small. The answer is false. | Yes |
| MCTS-$\alpha$ | Polly is a lepidopteran. Lepidopterans are insects. Polly is an insect. Every insect is an arthropod. Polly is an arthropod. Every arthropod is small. Polly is small. The answer is false. | Yes |
| MCTS-Rollout | Polly is a lepidopteran. Lepidopterans are insects. Polly is an insect. Every insect is an arthropod. Polly is an arthropod. Every arthropod is small. Polly is small. The answer is false. | Yes |

Table 23: Qualitative examples of ProntoQA outputs

| Prompt: What careers am I good at. | | |
| --- | --- | --- |
| Methods | Response | Reward |
| CoT | You can be a great career at any career you choose. Depending on your interests and skills, you may be able to find a variety of jobs that align with your interests and skills. | -2.625 |
| BFS-V | Some careers great careers to do professionally are entrepreneur in healthcare policy policy advocacy consulting public diplomacy leadership policy innovation in global issues managing information theory. Some great careers to achieve excellence in healthcare management include doctor psychologist | -2.03 |
| MCTS-$\alpha$ | You may be good at a variety of careers, depending on your interests and skills. Some of the most popular careers include software engineer, data scientist. You may also be good at a variety of fields such as finance, marketing, and business administration. | -1.67 |

Table 24: Qualitative examples of RLHF outputs

