# OpenReview forum: "Alphazero-like Tree-Search can guide large language model decoding and training"
_ICLR.cc/2024/Conference — Submitted to ICLR 2024_

### Official Review · Reviewer_1bFk · 2023-10-29

**Soundness:** 2 fair
**Presentation:** 2 fair
**Contribution:** 2 fair
**Rating:** 5
**Confidence:** 4

**Summary:**

This paper introduces TS-LLM, a tree-search learning framework tailored for Large Language Models (LLMs), drawing inspiration from AlphaZero. The framework sets itself apart from prior works by providing a unique method to enhance the performance and capabilities of LLMs in reasoning tasks and RLHF alignment.

**Strengths:**

- TS-LLM differs significantly from methods like Tree of Thought (TOT) and Reasoning via Planning (RAP), focusing less on reasoning tasks and not relying on large-scale models or human-designed prompts.
- The framework is adaptable for various tasks and LLM sizes.
- TS-LLM includes a learned value function and a final-step outcome reward model (ORM), enhancing performance in both decoding during inference and training.

**Weaknesses:**

Despite its merits, the paper does not demonstrate a significant improvement over previous works. Major concerns are:

1. The performance improvement is unclear and somewhat unconvincing.
   - **GSM8k:** The results are inferior compared to the previous work (ToT-BFS).
   - **Game24:** Despite MCTS-$\alpha$ winning, it requires **nearly twice** the number of tokens compared to other approaches.
   - **PrOntoQA:** Similarly, it requires **nearly twice** the number of tokens compared to other approaches.
   - **RLHF:** (See the next point)

2. Misapplication of RLHF Task: This paper's approach to the RLHF task seems misaligned with its intended purpose. Instead of tackling the fundamental challenges of RLHF, the study opts for a smaller agent (125M) and compensates with a higher token count to achieve elevated rewards through an open-source Reward Model. While this method may yield higher scores during training and potentially quicker convergence due to reduced sampling, the extended time required for "search" processes negates these benefits. Ultimately, this approach fails to address the crucial issues inherent to RLHF.

**Questions:**

1. In Table 4, why do DFS/BFS/MCTS yield identical results for the RLHF task? This paper should clarify this.

2. In Figure 1, the presentation should be further clarified. The authors did not evaluate Game24 on a token-level, however the figure may mislead readers.

3. Is the term "$k$-node" used in the context of BFS equivalent in meaning to "$k$-node" in the context of MCTS?

4. Do BFS and DFS utilize the same Large Language Model (LLM) as MCTS-$\alpha$ and MCTS-Rollout, or do they employ GPT-4 as their original Tree of Thought (ToT) configuration?

---

> ### Author Response · Authors · 2023-11-13
> **Response to Reviewer 1bFk (1/2)**
>
> We thank you for your valuable feedback and comments. We have modified several parts of our paper according to your suggestions.
>
> **Question 1: The performance improvement is unclear and somewhat unconvincing.**
>
> Thanks for your comment. We totally agree with your observation. There are three things we want to clarify:
> 1. **We want to clarify that TS-LLM’s DFS/BFS variants are different from that in Tree of Thought (ToT). They are TS-LLM’s variants instead of ToT baselines.** The main difference is that we use a learned value function instead of prompting advanced LLMs, which is one of our important contributions. We have shown in Table 4 the result of BFS and MCTS-alpha on Game24 by prompting LLaMA2-7B model as value function and ORM. It can exactly be regarded as an instance of ToT over small LLMs. Our conclusion is that prompting a small LLM can not offer accurate value estimation and the performance of tree-search is even worse than simple CoT baseline.
> 2. For path@1 results in Table 2, MCTS-alpha and MCTS-rollout do consume more tokens and it is reasonable to question whether it is mainly because of more token consumptions. Our Q4’s experiment can help to clarify such an issue. We additionally conduct experiments on search aggregation shown in Figure 3, where you can scale up your token consumption by running multiple times of search in MCTS/MCTS-alpha/MCTS-rollout/DFS or setting larger beam size in BFS. Specifically, the second row of Figure 3 can clearly demonstrate the performance of each tree-search algorithm when given the same token consumption. Basically, MCTS-alpha and MCTS-rollout are still dominant in most cases.
> 3. At last, we never claim in our paper that MCTS-alpha/rollout is the new SOTA over all evaluation tasks. Our experiments are not designed to show new SOTAs or the most efficient tree search algorithm. **We treat different search algorithms as TS-LLM’s variants and equally present their pros and cons. We believe every algorithm has its limitations and no one can claim superiority in all scenarios.** For example, as we illustrate in our analysis, MCTS, BFS, and DFS generally perform well in shallow search problems while MCTS-α and MCTS-Rollout are dominant in deep search problems.
>
> We thank you for your comment and understand that readers may have similar misunderstandings. To make this more clear, we’ve also modified this and incorporated all three points in our rebuttal revision. Specifically, we modify the Benchmark algorithms subsection in Section 4.1 to include more clarifications among benchmark algorithms and our experiment objectives. We’ve also modified the results analysis in Q1, to incorporate the Path@N result shown in Figure 3 for a better presentation.
>
> **Question 2: Misapplication of RLHF Task: This paper's approach to the RLHF task seems misaligned with its intended purpose. Instead of tackling the fundamental challenges of RLHF, the study opts for a smaller agent (125M) and compensates with a higher token count to achieve elevated rewards through an open-source Reward Model. While this method may yield higher scores during training and potentially quicker convergence due to reduced sampling, the extended time required for "search" processes negates these benefits. Ultimately, this approach fails to address the crucial issues inherent to RLHF.**
>
> Thank you for your comment, we fully agree with you that (1) we do not offer a solution that can solve the RLHF core problem like reward function learning (since we choose an open-source reward model) (2) The tree-search operations bring in additional computational burdens.
>
> Here what we are trying to provide is an alternative algorithm, in contrast to PPO or rejection sampling-based training, when given a fixed reward function. Since we are not a paper studying RLHF, this is just a showcase to present the applicability of the TS-LLM framework over tasks beyond reasoning. This is also a good example to show MCTS-alpha can do deep tree search in LLM generation (since we set a really wide and deep tree in the RLHF experiment.) In addition, there is one additional advantage of the tree-search algorithms. Section 3.2 has presented that we do not need to finetune the LLM generator using Rejection sampling finetuning or PPO, and can let the model generate reliable answers by pure inference.
>
> **Question 3: In Figure 1, the presentation should be further clarified. The authors did not evaluate Game24 on a token-level, however the figure may mislead readers.**
>
> Thanks for your comment, we have added the clarification in the footnote of Figure 1.
>
> **Question 4: Is the term k-node used in the context of BFS equivalent in meaning to k-node in the context of MCTS?**
>
> Exactly. The k is mentioned in the last paragraph of Section 3.1, as the number of nodes we subsample during the leaf node expansion. In other words, it is how many child nodes we have given a parent node.

---

> ### Author Response · Authors · 2023-11-13
> **Response to Reviewer 1bFk (2/2)**
>
> **Question 5: In Table 4, why do DFS/BFS/MCTS yield identical results for the RLHF task? This paper should clarify this.**
>
> Thanks for your comments and suggestions. We do clarify this in our paper in the last two sentences of the first paragraph in Q1’s answer.
>
> Let us clarify this more. When evaluating path@1 performance, we only generate one path given a search algorithm. It means MCTS cannot take the backward step (because it only conducts backward operation after it reaches the terminal node), BFS beam size is 1, and DFS cannot prune node and only get 1 path. In this case, MCTS, BFS, and DFS degenerate to greedy search over value. Results are different on the first 3 tasks because they are using sentence-level node expansion, so the action nodes are based on sampling. For the RLHF task, we derive the node by conducting a forward process and extract the top 50 tokens, it is deterministic as long as the input is the same.
>
> We fully agree that we need more explanations and have modified the analysis results in the first paragraph of our Q1’s answer. We’ve also clarified the Path@1 setting and additionally added a new result analysis by referring to the Path@N result shown in Figure 3. In this setting, MCTS, BFS and DFS will be different so we can conduct a better analysis. We’ve also added more explanations in the caption of Table 2 to better illustrate the reason why DFT/BFS/MCTS yield identical results.
>
> **Question 6: Do BFS and DFS utilize the same Large Language Model (LLM) as MCTS-alpha and MCTS-Rollout, or do they employ GPT-4 as their original Tree of Thought (ToT) configuration?**
>
> All search algorithms utilize the same LLM, which is LLaMA2-7B in the first three tasks and GPT-2 in the RLHF alignment task. We want to show that our framework is applicable to all different LLMs of any size and directly applying ToT pipeline to them may not work.

---

> > ### Author Response · Authors · 2023-11-17
> >
> > Dear reviewer, we appreciate your valuable feedback and constructive comments. Since the rebuttal process only has one week left, we would like to know whether our revision and rebuttal have addressed your concerns. We are open to engaging in further discussions and welcome any additional suggestions that can help enhance the quality of our paper.

---

> ### Comment · Reviewer_1bFk · 2023-11-21
> **Response to Question1&2**
>
> ### Question 1: The performance improvement is unclear and somewhat unconvincing.
>
> In your response to the question regarding performance improvement, you emphasize that your paper does not claim that MCTS-alpha/rollout establishes a new state-of-the-art (SOTA) across all evaluation tasks, and that your experiments were not designed to demonstrate new SOTAs or the most efficient tree search algorithm. While this clarification is appreciated, **it might be beneficial for the paper to more explicitly state the specific contexts or scenarios where MCTS-alpha and MCTS-Rollout demonstrate their strengths, beside from "going deeper".**
>
> considering the use of value networks in MCTS is **not a novel approach**, a deeper exploration of how your implementation improves upon existing models would further strengthen your argument.
>
> ### Question 2: Misapplication of RLHF Task
> If the primary goal is not to address the core problems of RLHF but to present an alternative algorithm, it might be more beneficial for readers if you explicitly compare your method with existing approaches like PPO and rejection sampling-based training. This comparison would help in understanding the unique advantages or limitations of your method in the RLHF domain.
> Moreover, while using an open-source reward model simplifies the process, it might also be worthwhile to discuss how this choice influences the generalizability and applicability of your findings. If your method shows promise in specific scenarios or configurations, clarifying these conditions would add valuable context for your readers, helping them understand where your approach fits in the broader landscape of RLHF solutions.

---

> ### Comment · Reviewer_1bFk · 2023-11-21
> **Other Comments**
>
> ## Clarity on BFS and MCTS Variants
> In your text, you replaced "BFS" with "BFS-V," suggesting a variant. However, this change might not be clear to readers that it is equivalent to Beam Search. For greater precision and clarity, changing the term to "Beam Search" would be more appropriate and straightforward. This adjustment ensures that the concept is immediately recognizable and accurately represented.
>
> Similarly, the use of MCTS needs clarification. If the paper employs a variant of MCTS from *Sampled MuZero,* mention this earlier. This distinction is crucial to avoid confusion, especially since it differs not only from traditional MCTS but also from the MCTS used in "AlphaZero."
>
> ## Fairness in Baseline Comparison
> Your paper uses CoT-greedy as a baseline, which may not provide a fair comparison with your methods. If your research builds upon ToT and RAP, it's essential to include and compare their results as well. While it's noted that these studies use GPT-4 and your method employs a smaller model, this contrast is precisely what readers need to understand. It's crucial to demonstrate why your method is preferable, outlining **any trade-offs or improvements they might expect.**

---

> ### Author Response · Authors · 2023-11-21
> **Response to Reviewer 1bFk [1/2]**
>
> We thank you for your valuable feedback and comments.
>
> **Question 1: While this clarification is appreciated, it might be beneficial for the paper to more explicitly state the specific contexts or scenarios where MCTS-alpha and MCTS-Rollout demonstrate their strengths, beside from "going deeper". considering the use of value networks in MCTS is not a novel approach, a deeper exploration of how your implementation improves upon existing models would further strengthen your argument.**
>
> We do explicitly state the strengths of MCTS-alpha and MCTS-Rollout in our analysis of Q1 and mention that these two algorithm variants dominate when dealing with deep-search problems while the rest three perform quite well in shallow one.
>
> Could you elaborate more on ' implementation improves upon existing models '? Or do you mean existing algorithms such as TOT and RAP? If so, please check the following answer to Question 5.
>
> **Question 2: Misapplication of RLHF Task**
>
> We do explicitly compare our methods to existing approaches like PPO and rejection sampling-based training. That is mainly in Q5 and Table 6 when we discuss how such a process can guide training.
>
> Thank you for your suggestion, we will clarify more about the choice of the reward model and further clarify the specific configuration.
>
>
> **Question 3: In your text, you replaced "BFS" with "BFS-V," suggesting a variant. However, this change might not be clear to readers that it is equivalent to Beam Search. For greater precision and clarity, changing the term to "Beam Search" would be more appropriate and straightforward.**
>
> Thank you for your feedback and suggestions. For the name of BFS-V, it doesn’t mean it’s just a variant. As we mentioned in Section 3.2.2, page 5, it means **“with value function based tree-pruning”**, suggesting the pruning is done by a learned value function which is different from Tree of Thought (ToT). We do agree with you that it can be called “Beam Search”, however, we maintained the name “BFS” to keep consistency with Tree of Thought paper. And then we added a suffix “-V” to show the main difference between the BFS(Beam Search) algorithm in TSLLM and Tree of Thought. To make it more clear, we also say in the main paper that “BFS-V can be regarded as a beam-search with cumulative reward as the objective” in Section 3.2.2, page 5.

---

> > ### Author Response · Authors · 2023-11-21
> > **Response to Reviewer 1bFk [2/2]**
> >
> > **Question 4: The use of MCTS needs clarification. If the paper employs a variant of MCTS from Sampled MuZero, mention this earlier. This distinction is crucial to avoid confusion, especially since it differs not only from traditional MCTS but also from the MCTS used in "AlphaZero."**
> >
> > Thank you for your feedback and suggestions. “Sampled Muzero” is irrelevant to a specific tree-search algorithm but relevant to the node expansion process. As we mentioned in Section 3.1, page 3, we refer to “Sampled MuZero” since when expanding the tree with **sentence-level action nodes**, the possible action space is the total language space, which is intractable, we use the LLM as a policy to sample w (tree-max-width) actions which is similar to the technique used in  “Sampled MuZero”.
> > We also think there should be more clarification on the different Monte-Carlo Tree Seach variants (MCTS-alpha, MCTS-Rollout and MCTS). We’ve added a comparison of each search variant in Appendix C.2.
> >
> > **Question 5: Your paper uses CoT-greedy as a baseline, which may not provide a fair comparison with your methods. If your research builds upon ToT and RAP, it's essential to include and compare their results as well.**
> >
> > Thank you for your feedback and suggestions.
> > We do conduct an experiment to see the performance of ToT/RAP on relatively small models(LLaMA2-7b). And we also emphasize they are the ToT and RAP setting.  We show the results in Table 4, Page 8, by letting the LLM output evaluations of the intermediate states themselves. It turns out they completely cannot work and even perform worse than a simple CoT-greedy baseline. We conclude that small LLMs might not be reliable value functions and evaluators.
> >
> > Moreover, we also care greatly about fairly comparing the performance between the linear decoding methods( CoT, CoT-SC) and tree search methods. Therefore, we also compare each method’s computation consumption by the number of generated tokens as we mentioned in Section 3.4. We also want to clarify that we do not use CoT-greedy as the only baseline which may not be a fair comparison. We introduce CoT-SC and conduct both **majority-vote** and **vote by aggregating with the outcome reward model(ORM)** as our baseline. The latter one, CoT-SC_ORM is a strong baseline, it outperforms CoT-greedy and CoT-SC majority-vote a lot as shown in Table 2. By taking comparisons of all different methods as fairly as possible, we verified our method is a generally applicable tree search framework from the perspective of performance, computation consumption, and continuous self-improving.
> >
> >
> > ---
> >
> > We hope our answers and results above have addressed your concerns. If so, we would greatly appreciate it if you could consider raising the score. Please let us know if there are any other questions.

---

### Official Review · Reviewer_xk9Q · 2023-10-31

**Soundness:** 2 fair
**Presentation:** 2 fair
**Contribution:** 2 fair
**Rating:** 5
**Confidence:** 4

**Summary:**

This paper proposes a method to use tree search algorithms to improve the decoding capability of LLMs. There are a couple of existing methods that have been applying tree search to improve the reasoning and planning capability of LLMs. For instance, Tree-of-thought (ToT) uses depth/breadth-first search to improve the planning capability of LLMs. However, the paper points out there are two major drawbacks in the prior work. First, the current methods mainly focus on enhancing LLM's reasoning ability. It is unclear if these methods can be applied to different kinds of tasks such as RLHF. Second, the existing methods heavily rely on hand-engineering prompts. As a result, such algorithms lack their applicability and heavily rely on both well-designed prompts and large-scale LM. As a result, the paper proposes a method that is based on MCTS. The main algorithm consists of two major steps: (1) policy improvement: it first generates tree-search trajectories based on the LLM, the value function that predicts the returns, and the reward model that predicts the end return; and (2) policy evaluation: based on the generated dataset from policy improvement, it trains the value function and reward model to further improve the performance. Several experiments are shown in section 4. Table 2 shows the mixed signal of whether the proposed method, MCTS, is better than the existing approach or not. Table 3 shows an ablation study of the MCTS model in Game24 tasks with a different allowable computation budget. Finally, the paper concludes by saying that it proposes a new training paradigm.

**Strengths:**

1. The idea of adding a value function and a reward function to further improve the decoding capability of LLM is a natural extension of the direction of applying a traditional planning algorithm to LLMs.
2. The experiments are very extensive, and the writing about the proposed method is easy to understand.

**Weaknesses:**

1. Based on the result in Table 2, it is unclear if the proposed method is better than the existing method such as BFS (ToT). For instance, in GSM8K, the BFS is better than the proposed methods such as MCTS-\alpha and MCTS-Rollout. We spent a good amount of time getting value functions and the reward model working, but the performance of the proposed method is still worse than the simple baseline. As a result, I am not convinced that this approach will work better or not.
2. Second, it seems that most of the experiments are about improving the value function and reward function, not about fine-turning the LLM. For example, if we do fine-tuning Table 2, are we able to improve the performance? Essentially, I want to see more results based on the question 5 written in the paper: Can TS-LLM further train LLM? I am happy to increase the score if the answers are answered.

**Questions:**

See the weakness section

---

> ### Author Response · Authors · 2023-11-13
> **Response to Reviewer xk9Q**
>
> We thank you for your valuable feedback and comments. We have modified several parts in our paper according to your suggestions.
>
> **Question 1: Based on the result in Table 2, it is unclear if the proposed method is better than the existing method such as BFS (ToT). For instance, in GSM8K, the BFS is better than the proposed methods such as MCTS-\alpha and MCTS-Rollout… As a result, I am not convinced that this approach will work better or not.**
>
> Thank you for your comment.  There are three things we want to clarify:
> 1. **We want to clarify that TS-LLM’s DFS/BFS variants are different from that in Tree of Thought (ToT). They are TS-LLM’s variants instead of ToT baselines.** The main difference is that we use a learned value function instead of prompting advanced LLMs, which is one of our important contributions. We have shown in Table 4 the result of BFS and MCTS-alpha on Game24 by prompting LLaMA2-7B model as value function and ORM. It can exactly be regarded as an instance of ToT over small LLMs. Our conclusion is that prompting a small LLM can not offer accurate value estimation and the performance of tree-search is even worse than simple CoT baseline.
> 2. Table 2 represents path@1 results. A much more comprehensive result can be seen in our Q4’s experiment. We additionally conduct experiments on search aggregation shown in Figure 3, where you can get Path@N results and scale up your token consumption by running multiple times of search in MCTS/MCTS-alpha/MCTS-rollout/DFS or set larger beam size in BFS. Specifically, the bottom line of Figure 3 can clearly demonstrate the performance of each tree-search algorithm when given the same token consumption. Basically, MCTS-alpha and MCTS-rollout are still dominant in most cases.
> 3. At last, we never claim in our paper that MCTS-alpha/rollout is the new SOTA over all evaluation tasks. Our experiments are not designed to show new SOTAs or the most efficient tree search algorithm. **We treat different search algorithms as TS-LLM’s variants and equally present their pros and cons. We believe every algorithm has its limitations and no one can claim superiority in all scenarios.**
>
> To make this more clear, we’ve also modified this and incorporated all three points in our rebuttal revision. Specifically, we modify the Benchmark algorithms subsection in Section 4.1 to include more clarifications among benchmark algorithms and our experiment objectives. We’ve also modified the results analysis in Q1, to incorporate the Path@N result shown in Figure 3 for a better presentation.
>
> **Question 2: Second, it seems that most of the experiments are about improving the value function and reward function, not about fine-turning the LLM. For example, if we do fine-tuning Table 2, are we able to improve the performance? Essentially, I want to see more results based on question 5 written in the paper: Can TS-LLM further train LLM?**
>
> We do present experimental results by finetuning Table 2 and Figure 3. The specific results are presented in Table 5 and Table 6. In Table 5, we present how we can leverage tree-search examples in further finetuning the LLM-based value function. In Table 6, we show more comprehensive results over GSM8K and RLHF alignment and present how tree-search examples can increase LLM generation by finetuning LLM policy $\pi_{\theta_0}$ to $\pi_{\theta_1}$, LLM value $V_{\theta_0}$ to $V_{\theta_1}$ and both. Does this address your concerns?

---

> > ### Author Response · Authors · 2023-11-17
> >
> > Dear reviewer, we appreciate your valuable feedback and constructive comments. Since the rebuttal process only has one week left, we would like to know whether our revision and rebuttal have addressed your concerns. We are open to engaging in further discussions and welcome any additional suggestions that can help enhance the quality of our paper.

---

> > > ### Author Response · Authors · 2023-11-22
> > >
> > > Dear Reviewer xk9Q, we hope our answers and results above have addressed your concerns. If so, we would greatly appreciate it if you could reconsider your score. Please let us know if there are any other questions.

---

### Official Review · Reviewer_E6Fb · 2023-11-05

**Soundness:** 2 fair
**Presentation:** 3 good
**Contribution:** 2 fair
**Rating:** 6
**Confidence:** 4

**Summary:**

The paper proposes a set of MCTS-based algorithms for language generation using LLM that can replace beam search or further fine-tune the LLM. The general idea is to train a value/reward model based on ground truth correctness for reasoning task or rewards in datasets used for RLHF. By unifying the notion of desire paths in a flexible model, the implicit policy of vanilla LLM decoding can be combined with exploration or valuations of incomplete states, so heuristic-search algorithms like MCTS become feasible.

The paper considers token and sentence action spaces. The resulting set of decoding can be aggregated in multiple ways.

The experiments use zero-shot Chain-of-Thought (CoT) via fine-tuning as a baseline. The datasets include some common reasoning ones, a new one in reasoning and another one on RLHF.

For a more fair comparison, the results aim to compare algorithms under similar amounts of tokens generated, allowing COT to generate further paths for posterior aggregation.

Under the specific hyper-parameters used, the empirical results show MCTS as a promising direction.

**Strengths:**

- Separation of concerns between the LLM as a selection of action/tokens vs the value of a state that can integrate information about past sequences and possible continuations.
- Multiple algorithms cover most of the meaningful combinations.
- Attempt to fair comparison on the effort to the number of tokens
- Diverse set of datasets
- Systematic combination of algorithms and settings
- Some discussion on the challenges of the implementation

**Weaknesses:**

- No details on the value/reward model besides the mention that it is a decoder-only model.
- Explanation of the algorithms doesn’t clarify when the value/reward models are used.
	- Some details are buried in the appendix but need to be clearer.
	- In particular, knowing how the evaluation function propagates information for training and during search is critical.
- Some settings need to be fully explained and unclear what version was used.
	- For instance, the appendix says that sometimes intra or inter-tree search is used, but it’s not clear which experiment uses which setting.
	- Part of that can be implied from the algorithm descriptions, but that’s not the reader’s job.
- No discussion on how the hyper-parameters were set.
	- This is especially critical for MCTS, as it might be very sensitive to them.
	- Many other details, such as the 0.3 in the Diriletch distribution for adding noise, are important.
- No specific statistics about the behaviour of the algorithms beyond the aggregated number of tokens.
	- For instance, knowing the number of roll-outs would be useful.
- No discussion of the wall-time for decoding
	- While the paper discussed implementations, it’s not clear what the actual performance might be. While the implementation can be challenging and improved if this method becomes more popular, we need to know about the current cost of solving the tasks.
	- For instance, keeping the tree in GPU memory can be expensive. Moving tensors in and out could reduce GPU utilization.
- Analysis can be improved
	- For instance: *Highlight [page 8]:* The results of CoT-SC ORM-vote@10 underscore the diversity of sampled data in learning a better ORM.
	- But this is not affecting the MCTS ones, so the claim about the diversity is only for the ORM.


Given the lack of details on how the hyper-parameters were set, and how sensitive the results are to them, it’s hard to qualify the contribution, and how likely these algorithms are to perform well in other domains. This perception can change depending on the answer of the authors.

Other issues:
- The notion of aggregation is misleading (Sect 3.2.3)
	- Aggregation refers to using a set of things to get something new, but in this case, we are just selecting an output, not combining them.
- The most relevant work is in the appendix:  *Highlight [page 13]:* The most relevant work, CoRe (Zhu et al., 2022), proposes to finetune both the reasoning step generator and learned verifier for solving math word problems using MCTS for reasoning decoding which is the most relevant work to ours.
	- Please fix the reference to ACL 2023.
- Make sure all references are used in the paper.
	- For instance, Self-consistency (4.1) doesn’t cite the reference already in the paper.

Suggestion / minor issues
- Given that MCTS combines both the policy and the value, perhaps a comparison with BFS or DFS is not the more meaningful. Algorithms based on A* rely on the accumulated cost g —in this case inverse to the likelihood of the prefix— and h, an heuristic estimating the value of the rest of path. While the value function in the paper does not take into account the length, it’s still possible to consider their combination.
- Please define the state $s$ by itself, not inside $g$:
	- *Highlight [page 4]:* given state s and action a, is known as: g ( s = (x 0:L−1, y 0:t−1), a = y t) = (x 0:L−1, y 0:t).
- Why defining $g$ in page 4? Section 3.1 uses \phi_theta
- *Highlight [page 6]:* This helps us juxtapose the performance
	- Juxtapose? Perhaps it’s a meaningful comparison
- *Highlight [page 7]:* For value and ORM training, the data are generated by randomly sampling the SFT policy’s rollouts on the training set
	- Please define SFT. I guess it’s supervised fine-tuning.
- Section “Background of Monte Carlo Tree-search Algorihtms” should say which algorithm is closer to alpha-zero.

**Questions:**

- Is there any restriction about the context size of the LLM and the value/reward models?
- Was any fine-tuning or training using multiple-GPUS or the A800 were used to run independent experiments?
- Value and reward models
	- Is this a single decoder-only model that produces the intermediary values and the final rewards, or are they different models?
	- If they are different models, can you report on the difference between v^_theta() vs r^_theta() when evaluated in final states?
- hyper-parameters:
	- How were the hyper-parameters setup, from Sect 4.1, task setup until the constants for MCTS.
	- How much budget was used to set these numbers?
	- How sensitive are the results to the particular parameters?
- In general, how hard should it be to adapt the proposed methods to another domain?
- Is BFS taking into account the likelihood of the previous path? The text only says to keep the ones with maximal value.
	- (Suggestion: see comment about A* above)
	- If it completely ignores the likelihood, then why does the decoding do something useful?
- What are the computational requirements for only training the value/reward models without fine-tuning?
- What is the distribution of wall-time per algorithm and dataset?
	- Section “Limitation and future work” discuss how limited is the implementation but it’d still be informative to know what’s the current status
- Are there any meaningful combinations that were not tested due to complexity?
	- Section D includes details but it’s not clear if all the challenges were overcome or some meaningful experiments might be missing.
	- This answer should be included in the discussion, or at least in appendix B.
- Were the samples in the dataset always selected from the train set?
- Is Path@N counting the number of paths up to a leave node or the number of separated paths?
	- For instance, for beam-search with size K, the number of leaves is at most K, but there were other paths partially explored that were discarded along the way.
- Path@N for CoT-SC variants
	- *Highlight [page 7]:* For CoTSC variants, we present Path@N, and N is determined by maintaining the same level of token computation as TS-LLM’s variants
	- How is the same level of token computation maintained? For instance, it could be the maximum average of the tokens used for all the variations of MCTS. Or perhaps, it was run with multiple values of N, and then return bigger N under the token of the other method. But I’d expect that to be done per sample.

---

> ### Author Response · Authors · 2023-11-13
> **Response to Reviewer E6Fb (1/3)**
>
> We really thank you for your valuable feedback and helpful suggestions. Here’s our response to your feedbacks and questions.
>
> **Question 0: Writing issues and more clarifications for TS-LLM.**
>
> According to your suggestion, we’ve made the following modifications and clarifications to our paper to address the writing issue you mentioned:
> 1. We’ve corrected the wrong version of the citation of CoRe and cited the self-consistency paper in Section 4.1.
> 2. We’ve modified the description of MCTS variants in the paper in Section 3.2.2 and added a comparison of the tree-search algorithms in TS-LLM in Appendix C.2.
> 3. We’ve changed the “juxtapose” on page 6 Section 3.4 into a more precise expression.
> 4. We’ve defined the state $s$ in Section 3.2.1 outside the transition function $g$.
> 5. We’ve defined SFT as “supervised finetuning” in Section 4.1.
> 6. We’ve modified the analysis to make it more clear and precise in Q3’s analysis in Section 4.2.
> We’ve added the necessary clarification about whether intra- or inter-tree search is used in Table 2 and Table 3 and clarify the detailed setups in Appendix D.5.
>
> **Question 1: The notion of aggregation is misleading (Sect 3.2.3). Aggregation refers to using a set of things to get something new, but in this case, we are just selecting an output, not combining them.**
>
> We totally understand your concern that the notion of ‘aggregation’ might be misleading. While we adopted it from the previous papers CoT-SC(https://arxiv.org/abs/2203.11171) and   RAP(https://arxiv.org/abs/2305.14992) for consistent terminology.
>
>
> **Question 2: No details on the value/reward model besides the mention that it is a decoder-only model.**
>
> As we mentioned in section 3.2.1, the value/reward model architecture is simple, we append a value head (which is an MLP that outputs a scalar, e.g. a linear layer) to a decoder-only transformer. For GSM8k, Game24, and ProntoQA, the decoder-only transformer is the pretrained LLaMA2-7b without the `lm_head` module. For the alignment task, the decoder-only transformer is the pretrained GPT2-small without the same output `lm_head`. And we used the pretrained weights as the initialization.
>
>
> **Question 3: No specific statistics about the behaviour of the algorithms beyond the aggregated number of tokens. For instance, knowing the number of roll-outs would be useful.**
>
> Thank you for your comment. We want to confirm the meaning of ‘roll-outs’ here. If it means the number of sequences a certain search algorithm outputs, we’ve already shown it in the first row of Figure 3 and Appendix D.6. We hope for your further explanation about the definition of ‘roll-outs’ and we will try to show you the results.
>
>
> **Question 4: No discussion of the wall-time for decoding**
>
> Thank you for your comment. We’ve added different algorithms’ decoding wall-time of GSM8k, Game24 and ProntoQA in Table 13~15 in Appendix D.9. The results demonstrated the gap of computational efficiency between CoT/CoT-SC decoding and tree-search algorithms in TS-LLM, due to the complicated search procedures and extra computation introduced by calling value functions in the intermediate states.
>
> We totally agree that computational efficiency is an important issue and address it in Appendix B. But note that our current implementation is just an algorithm prototype without any engineering acceleration. We are continuously working on accelerating our tree search framework codebase including caching past key/values on the tree to eliminate repeated computations and offloading tensors on RAM, etc.
>
>
> **Question 5: Given that MCTS combines both the policy and the value, perhaps a comparison with BFS or DFS is not the more meaningful. Algorithms based on A\* rely on the accumulated cost g —in this case inverse to the likelihood of the prefix— and h, an heuristic estimating the value of the rest of path. While the value function in the paper does not take into account the length, it’s still possible to consider their combination.**
>
> Yes, we agree with your ideas. This can actually be viewed as a dense reward setting in which, at each step, there is a nontrivial reward. It is natural and heartening to extend our method to this setting like extending from AlphaZero(https://arxiv.org/pdf/1712.01815.pdf) to MuZero(https://www.nature.com/articles/s41586-020-03051-4).
>
> **Question 6: Is BFS taking into account the likelihood of the previous path? The text only says to keep the ones with maximal value. If it completely ignores the likelihood, then why does the decoding do something useful?**
>
> No, BFS can be viewed as beam search with our learned value and ORM. However, the likelihood is not totally ignored. It will implicitly influence the search process since the actions proposed by LLMs are sampled w.r.t. the likelihood and then treated as i.i.d. sentences/actions in BFS. The decoding is useful since BFS will leverage the value function to choose the top K sentences/actions at each step.

---

> ### Author Response · Authors · 2023-11-13
> **Response to Reviewer E6Fb (2/3)**
>
> **Question 7: Why defining $g$ in page 4? Section 3.1 uses \phi_theta**
>
> We guess you mean $\pi_\theta$ in Section 3.1. They have different meanings here, $\pi_\theta$ is the policy that takes an input $x$ and outputs actions. While $g$ is defined in the dynamics transition function of the problem, for the tasks we are testing now the transition function is deterministic, i.e. appending output tokens to the prefix/prompt string. The transition function $g$ can be considered being influenced by the LLM policy $pi_\theta$ as $g(s, a) = g(s, a \sim \pi_\theta(\cdot | s))$
>
> **Question 8: Is there any restriction about the context size of the LLM and the value/reward models?**
>
> The context size of the LLM and value/reward models are restricted by the pretrained length of the LLM and the decoder-only transformer in the value/reward model.
>
> **Question 9: Was any fine-tuning or training using multiple-GPUS or the A800 were used to run independent experiments?**
>
> As we mentioned in Appendix D.2, the finetuning and training were conducted on 8 NVIDIA A800 GPUs.
>
> **Question 10: Is this a single decoder-only model that produces the intermediary values and the final rewards, or are they different models?**
>
> Yes, we use one decoder-only transformer model with a value head to output scalars at each token.
>
> **Question 11: How were the hyper-parameters setup. How much budget was used to set these numbers? How sensitive are the results to the particular parameters?**
>
> During our experiments, we found that the performance and token consumption are sensitive to several hyperparameters: first is the general hyperparameters of building the search trees, e.g. tree-max-depth and tree-max-width, for the former, we choose the max-depth according to the statistics of the distribution of the number of steps from the dataset sampled by the LLM policy on the training set; while for the latter we’ve shown in Table 3 that there is a trade-off between token consumption and the size of the search space.
> MCTS variants (MCTS-alpha, MCTS-rollout and MCTS) are also sensitive to hyperparameters, we keep most hyperparameters as the default values used in AlphaZero and MuZero except for $c_{init}$ which we select a value from two possible ones {0.3, 3.0} when computing $c_{puct}$ to control exploration and exploitation in MCTS variants.
> For the Dirichlet noise, this might be important since there is some guess that DeepMind sets this hyperparameter according to the statistics of the dataset collected for each task. However, due to the limit of computation resources, we just adopted the default value in AlphaGo as 0.3, which is specified for chess (refer to Page 14 in the AlphaZero paper (https://arxiv.org/pdf/1712.01815.pdf) and Appendix B in MuZero(https://www.nature.com/articles/s41586-020-03051-4)).

---

> ### Author Response · Authors · 2023-11-13
> **Response to Reviewer E6Fb (3/3)**
>
> **Question 12: In general, how hard should it be to adapt the proposed methods to another domain?**
>
> One of the contribution of TS-LLM is that it is generally applicable. The minimal requirement of our proposed framework is a set of problems/tasks and an oracle reward function(maybe rule-based) on them. Therefore,  this requirement can be satisfied for a wide range of NLP tasks and decision-making tasks.
>
> **Question 13: What are the computational requirements for only training the value/reward models without fine-tuning?**
>
> First, we need to do inference on the training problem set with the LLM policy to sample enough data for value/reward training. The value/reward model are then trained on such data. Since the network architecture of the value/reward model we used was almost the same as the LLM policy, finetuning such a model has a similar computational requirement as finetuning an LLM. Although we did update all weights of the value/reward model which have the same size as the LLM policy which indeed requires advanced GPUs like A800, for users with limited computational resources, using a smaller decoder-only transformer or techniques like LoRA would be an efficient and effective solution.
>
> **Question 14: Are there any meaningful combinations that were not tested due to complexity?**
>
> There are a lot of potentials to try and we have listed some in Appendix B discussing limitations and future work. For example, more complex tree construction strategies such as mixed sentence/token-level node expansion, adaptive node-max-width w.r.t. depth, and rejecting semantically similar action nodes to increase search efficiency and diversity.
>
> **Question 15: Were the samples in the dataset always selected from the train set?**
>
> Yes. For the finetuning of both the LLM policy and the value/reward model, we construct the dataset from the samples in the training set.
>
> **Question 16: Is Path@N counting the number of paths up to a leave node or the number of separated paths? For instance, for beam-search with size K, the number of leaves is at most K, but there were other paths partially explored that were discarded along the way.**
>
> For BFS(beam-search) and DFS, the answer is the number of separated paths. For MCTS variants, this means running the search algorithm N times, they are not ensured to be distinct.
>
> **Question 17: How is the same level of token computation maintained? For instance, it could be the maximum average of the tokens used for all the variations of MCTS. Or perhaps, it was run with multiple values of N, and then return bigger N under the token of the other method. But I’d expect that to be done per sample.**
>
> We just choose N which is close to the most tokens used in all tree search algorithms. We want to clarify here such a process is reasonable. We’ve counted the number of tokens for the CoT outputs of each task, we found that the number of tokens among every problem in all tasks has small standard deviations compared to the average values (e.g. for GSM8k it’s $101 \pm 36$, for Game24 it’s $79 \pm 3$, for ProntoQA it’s $92 \pm 17$).

---

> ### Comment · Reviewer_E6Fb · 2023-11-15
> **Thank you**
>
> Thank you for your comments.
>
> On BFS: since this is not using the LLM likelihood at all, the algorithm seems to be out of the scope of the paper. Right?
> If you want to keep it, this needs further justification.
>
> Instead of BFS, the natural algorithm to compare is A*. For DFS, the alternative would be IDA*, which might be tricky to implement.
> I think that A* is the baseline for comparison, no BFS.
>
> A* uses the accumulated cost (cost is inverse to the accumulated likelihood) and an estimation of the future.
>
> Regarding the roll-out, we generally want to understand the search effort besides the implementation detail. A typical metric in search is the number of expansions: how many LLMs calls. Another one is the number of calls to the other models. As the different algorithms use some models and not others, this would reinforce the explanation of the algorithm. The roll-outs may only be for the cases where the leaves are reached.
>
> These metrics are crucial to understand the trade-off of the algorithms.

---

> ### Author Response · Authors · 2023-11-16
> **Response to Reviewer E6FB**
>
> Really thank you for your feedback!
>
> For the BFS/DFS algorithm, we do not think it is out of the scope of our paper. Firstly, we want to clarify that, in our paper, DFS/BFS are TS-LLM’s variants instead of ToT baselines. We never claim in our paper that MCTS-alpha/rollout is the new SOTA over all evaluation tasks. We believe every algorithm has its limitations and no one can claim superiority in all scenarios. For example, as we illustrate in our analysis, BFS/DFS generally perform well in shallow search problems.
> Secondly, for MCTS/MCTS-alpha and MCTS-rollout, they need the LLM likelihood and LLM value function, while for DFS/BFS, they still need the LLM value function to help prune the tree (and implicitly rely on the log probability to expand nodes by sampling). Learning a value function is one of our important contributions and DFS/BFS can help us verify the quality of value function. Could you explain more on why it is necessary to explicitly include LLM likelihood in the search process?
>
> For the A* algorithm, we fully agree this is another search algorithm we should include in TS-LLM's variants. We are working on adding this algorithm, but it takes some time to rerun all our tasks and tune the search hyperparameters.
>
> For the rollout, the token number we monitor in our experiments (token number in Table 2 and the x-axis in the second row of Figure 3), can exactly represent the node expansion times. This is another contribution we have - we want to conduct a fair comparison (Section 3.4) which prior works neglect. We sum up all the new tokens generated by LLM every time the node expands. In addition, we think the statistic of the number of generated tokens is better than mere node expansion times. Note that it also takes the expense of each expansion into consideration since it is possible that different node expansions may generate different lengths of sentences (number of tokens), resulting in varying computation expenses.
>
> About another metric like the number of calls to other models, as we described in TS-LLM, we do have calls of the LLM value/reward model. But again, the computation expense of calling value/reward model after each expansion can be estimated accurately by the number of newly generated tokens.

---

> ### Comment · Reviewer_E6Fb · 2023-11-16
> **Good point**
>
> Good point. If somehow BFS was considering all the tokens, then it would not be using the LLM likelihoods. That BFS would be out of the scope of the paper. However, BFS must consider all the tokens.
>
> So, the algorithm called BFS in the submission is not BFS because it’s using the LLM likelihood to prune the tree. So it’s actually a two steps process: that might be mimicking A*: the token continuation discarded can be seen has having infinite cost g. The ones preserved have a finite cost depending on the depth. The models proposed in the submission are used to decide among the new nodes, such that older nodes are always expanded first.
> This can be exactly A* if we consider the cost g increasing by a number bigger that the maximum value that the models can provide. I think it’s, so just increasing g by 2 with each token or sentence (don’t remember now) should be enough to see the BFS of the paper as an instance of A*.
>
> So, I agree now that BFS is in a the scope of the paper. (Here is for me a demonstration of the value of an interactive discussion with the authors).
>
> However, the name BFS not right, even if the previous work using BFS for LLM use it. As for the aggregation, the fact a few papers of a few years ago used a terminology doesn’t mean that new papers should continue to pollute with confusion the archive of accepted papers. It’s totally ok to use a different term, as far as the manuscript mentions that previous recent work use another term.
>
> I suggest not using *aggregation* and I suggest that just calling this algorithm BFS leads to confusion. Perhaps it’s a matter of emphasizing that this is doing BFS in the space after pruning with the LLM.
>
> That view suggests that the results might be very different depending on the amount of pruning. No pruning doesn’t use the LLM. Prune greedily, keeping one solution is greedy search. In the middle, more or less pruning might lead to very different behaviour. Less pruning is like higher recall while BFS chooses among them.
>
> *New question*: what’s the fundamental difference between beam search and BFS? Perhaps beam search is just fixing how many solutions are preserved while BFS might take a different number of nodes at different points of the search. Perhaps flexible beam search would be a more precise name.
>
> *new question*: did you test different amount of pruning for BFS? Without such variation, this quote in the last comment doesn’t hold:
>
> > DFS/BFS can help us verify the quality of value function.
>
> Because there selection of the amount of pruning might depend on the domain, and finding a the magic amount of pruning be non trivial, especially if going off distribution during testing.
>
> Let’s call Oracle-BFS the one using the optimal pruning by LLM for producing the result. The performance of Oracle-BFS could dominate in some domains variations ot MCTS that doesn’t have such oracle and doesn’t commit to a specific amount of pruning. On the contrary, BFS as in the paper  might not say much because perhaps the pruning is close to optimal or suboptimal. The demonstration is interesting but I don’t think it’s more than a demonstration.
>
> I remain open to the possibility that further explanation on the pruning might convince me otherwise.
>
> Thanks

---

> > ### Comment · Reviewer_E6Fb · 2023-11-16
> > **Got It**
> >
> > Ok. It was right there in the manuscript:
> >
> > > BFS can be regarded as a beam-search with cumulative reward as the objective. In BFS, nodes with k-largest values are maintained and expanded in each step.
> >
> > So, perhaps BFS can be the name, but the significance depends on the value k more heavily than other algorithms.
> >
> > *question*: what is the k used for BFS? Is it the Tree Max Width in table 1?
> >
> > *question*: how sensitive are the results to the selection of k for BFS and related parameters in the other algorithms?
> >  (this might be have been answered  above but it’s a good recap)
> > An algorithm more robust to k is more promising for new domains.

---

> > > ### Comment · Reviewer_E6Fb · 2023-11-16
> > >
> > > Regarding my last question: path@1 is reported as equivalent to greedy search but for path@N -when applies- and the default setting is less clear.
> > >
> > > The comment above says:
> > >
> > > > while for [tree max width] we’ve shown in Table 3 that there is a trade-off between token consumption and the size of the search space.
> > >
> > > Table 3  shows variations for Game24 but not for the others.
> > >
> > > **KEY POINT**:
> > >
> > > **The so-called “node construction” studied in Question 2 or the paper might be a very hard to tune parameter. Therefore, the value of the contribution might be tainted as finding the right number might be very hard. ideally, we should see tables like table 3 but for the other two domains.**
> > >
> > > So my question about the sensibility becomes more critical. For instance, Table 3 suggests that if pruning of aggressive (only 6), then perhaps BFS would be simpler, cheaper and better.
> > >
> > > I’m really curious about how the max width was set. Even looking at the dataset is itself problematic as a the method would depend on expertise about the domain.
> > >
> > > I’m afraid I’m a bit concerned about this issue and it might affect my scores negative. I hope this discussing dissipate my increasing doubts.
> > >
> > > I’ll stop for now. Two last comments:
> > > * what about calling it “reward beam search” instead of BFS?
> > > - the term “node construction” is also confusing. I understand the engineering motivation but it should be called something like pruning or selection. Expansion or prediction might work. Otherwise, I’d check the terminology for the popular beam search that is usually more crisp.

---

> ### Author Response · Authors · 2023-11-16
> **Response to Reviewer E6Fb (1/2)**
>
> Thanks for your feedback and helpful suggestions. Here’s our response to your questions.
>
> **Question 1: What is the k used for BFS? Is it the Tree Max Width in table 1? Did you test different amount of pruning for BFS? How sensitive are the results to the selection of k for BFS and related parameters in the other algorithms?**
>
> Let us explain this more clearly. **Tree-max-width and k are two different hyperparameters. Tree Max Width determines the number of child nodes every time the tree expands while k represents the number of child nodes we maintain (based on top k value) from the whole candidates.** For example, assuming we maintain k nodes at depth d, when expanding nodes for depth d+1, we will expand these k nodes to have **(k \* tree-max-width) child nodes, from which we select the top k nodes. After this, we have k nodes at depth d+1 and the search continues**. The k can also be treated as beam size, which is a hyperparameter to control how many distinct paths are to be sampled on the tree in the BFS(Beam-Search) algorithm.
>
> And Yes, we do test it in Q1/Q4 and Figure 3. The beam size k controls the max number of returned sequences and it corresponds to $N$ in Path@N experiments, as is shown in the first row of Figure 3. To make the setting of Path@N results clearer, we’ve also added explanations about the detailed settings used in the aggregation experiments in Appendix D.9.
>
> About the sensitivity of k, our experiment in Figure 3 has shown that, in GSM8k/ProntoQA/Game24, the performance of BFS is proportional to the size of k (It is reasonable because we have an ORM to help rerank all final k solutions to determine a final solution, and larger k represent larger candidate space). RLHF experiment is a special case where we find BFS does not work in deep search problems. So if you ignore the extra computational burden brought by increasing k/N, in most cases, you can obtain better performance by using larger K. In this case, you won’t need to determine the specific k. However, with k growing, you also have to consider the extra computational burden because you need to expand/evaluate more nodes in BFS. So this is also a trade off between larger search candidates (larger k) and heavier computation demands.
>
>
> **Question 2: The sensitivity of hyperparameter such as how the max width was set.**
>
> We fully understand your concern about how sensitive are the results to the hyperparameters. This is exactly what we want to present in our paper. We want to present a comprehensive and systematic analysis about the key components of tree-search guided inference and training in LLMs including alternatives of tree-search algorithms, training of value function, aggregation (or maybe called reranking) of multiple searches, potentials to continuously improve the total framework, and the possible trade-off between different tree construction(expansion) settings (i.e. choices of sentence-/token-level action nodes,  the **tree-max-width** and tree-max-depth).
>
> We agree with you that tree construction hyperparameters might greatly influence the performance and computation cost of tree-search algorithms. Therefore, we conducted ablations on Game24, which is, among the four tasks tested in the experiments, a typical task to show **how the selection of tree-max-width could affect the search space upper bound and the final performance**. That’s because if one intermediate reasoning step (computation of two remaining numbers) is wrong (impossible to reach the final result of 24), the total subtree from this node is meaningless since all of them should be wrong answers. This phenomenon was also pointed out in the Tree of Thought paper that most failures often happened in the very early steps of Game24.
>
> Let us illustrate more on how we choose the hyperparameter of tree-max-width. For RLHF, we choose it as the default hyperparameter in LLM decoding( top_k=50). For GSM8K/Game 24/ProntoQA, we first experiment with a small tree-max-width and increase it to see the performance gain by increasing the search space upper bound. This is similar with k mentioned in Q1. The larger the search space, the better the performance. Finally, we determine the tree-max-width by fairly considering the tradeoff between the search space limit and computational burden (This is why in Table 3, width=50 is better than width=20 but we still choose width=20 in Table 1). The procedure we follow is a common way for most search algorithms when given a new environment and it is not hard to tune at all. You can always customize your tree search space based on your requirements for the performance and computation budget.

---

> ### Author Response · Authors · 2023-11-16
> **Response to Reviewer E6Fb (2/2)**
>
> **Question 3:  For instance, Table 3 suggests that if pruning of aggressive (only 6), then perhaps BFS would be simpler, cheaper and better.**
>
> Firstly, when we set the tree-max-width as 6, we are not conducting pruning, we are limiting the search space upper bound of all algorithms by limiting the number of child nodes an intermediate node can expand.  And it is totally fine that BFS in some cases is better than MCTS-alpha/MCTS-rollout. We have addressed this problem in our global response and our previous response to you. BFS is one of TS-LLM’s variants instead of merely a baseline. Specifically, we said:
>
> We never claim in our paper that MCTS-alpha/rollout is the new SOTA over all evaluation tasks. Our experiments are not designed to show new SOTAs or the most efficient tree search algorithm. We treat different search algorithms as TS-LLM’s variants and equally present their pros and cons. We believe every algorithm has its limitations and no one can claim superiority in all scenarios.
>
> And the final experimental results validate our belief. BFS/DFS/MCTS algorithms perform well in shallow search problem but struggle in deep search problem (RLHF).
>
>
> **Question 4: What about calling it “reward beam search” instead of BFS?**
>
> Thank you for your comment. We think reward beam search is also an alternative. But to align with previous papers such as Tree of Thought, we also think it’s reasonable to mention the name BFS. So to consider both, what do you think about terms like BFS-V/DFS-V (abbreviation for BFS/DFS with value pruning)?
>
>
> **Question 5: The term “node construction” is also confusing.**
>
> We understand your concerns. We have replaced the term with “node expansion” in Q2.

---

> > ### Comment · Reviewer_E6Fb · 2023-11-16
> > **Rest of the answers**
> >
> > Thank you for the prompt response and the clarifications.
> >
> > I didn’t acknowledge the rest of the answers. Thank you.
> >
> > I realized there was a mistake about the take-away from the work, but perhaps was a fair confusion. The work is not about combining extra models with an LLM. The paper says it clearly: this is inspired in AlphaZero and variants. So, it makes sense to use the same pretrained LLM, only adding a head for the value. This is not a limitation but doubling down on the AlphaZero idea where agents live forever in the same game so it makes sense to have a shared representation for the different models, as the models are specialized on a particular task.
> >
> > In the submission, “learning a value function” might not convey the right message, as *a value function* might as well be completely independent. It could be a decision tree over the state/text, for instance. Instead, this work is about a form of fine-tuning.
> >
> > This confusion was behind my original Q2, Q8, Q10 and Q13.
> >
> > **This must be clarified since the abstract: the new models are using the same pretrained LLM.**
> >
> > That can be in the text, but perhaps a small figure after Fig 1 can illustrate that the experiments are about a given pre-trained LLM is extended with another head, and how those heads are the functions used in the algorithms. It be even better if that figures says which algorithm uses which model.
> >
> > Regarding other responses, your answer to my original Q3 show there I was a confusion we elaborated in further comments.
> >
> > I’ll go back to k, N and expansion in the next comments.

---

> > > ### Comment · Reviewer_E6Fb · 2023-11-16
> > > **Terminology**
> > >
> > > In the body of the paper, it is not important to follow the terminology used in a recent paper used, not matter how influential. I understand using those terms in the abstract, while keeping the interpretation open to adjust in the introduction. But from the introduction is better to be precise, and the fast-pace of publication is not helping with that.
> > >
> > > * I’m ok with BFS-V. At least it invite the reader to understand what it is, instead of assuming that the algorithm is standard BFS.
> > > * Node expansion is a good choice as the terminology is used for describing MCTS.
> > > * Aggregation vs selection.

---

> > > > ### Comment · Reviewer_E6Fb · 2023-11-16
> > > > **Remove inter-tree for a clear state space?**
> > > >
> > > > For this paper, it doesn’t make sense to look for SOTA, I agree. Instead, the attempt to do a systematic evaluation is much better. However, the systematic evaluation of search algorithms only makes sense in the same state space.
> > > >
> > > > Inter-tree is not looking at the same state space, it can be more expensive, and if resampling was crucial, then we might want to revisit all algorithms to allow resampling. For instance, an extension of the other TS algorithms that resamples part of a past beam, equivalent to a *restart* as commonly done in SAT solvers and in some combinatorial optimization search algorithms.
> > > >
> > > > I suggest to move *inter-tree algorithms* to
> > > > * another question in Section 4.2
> > > > * or move it all together to the appendix.
> > > >
> > > > I prefer the latter, as it would allow us to
> > > > * reduce the number of configurations presented in the paper.
> > > > * revise the problem formulation, section 3.1, to end by clarifying that in this paper the the policy is deterministic once the LLM prediction is performed, whether a token level or a sentence level, constituting a proper state space.
> > > >
> > > > So far it’s mostly adding confusion. That’s almost an idea to be fully studied in a follow-up paper. Restart in combinatorial optimization helps to alleviate the issue of “early commitment”, as an early decision might lead to not finding good solutions. In this case, resembling might be a way to escape a region full of bad predictions.
> > > >
> > > > Moving away inter-tree to results or appendix would simplify the discussion around tree width, k, N, and expansions.
> > > >
> > > > In that regard, the variable $k$ is used with multiple meanings:
> > > >
> > > > * In Sect 3.2.2, for MCTS-rollout: “k complete answers”, for BFS: “k largest values”.
> > > > * A comment above says that k is N, at least for BFS:
> > > >
> > > > > The beam size k controls the max number of returned sequences and it corresponds to  in Path@N experiments, as is shown in the first row of Figure 3.
> > > >
> > > > The record in the literature doesn’t help: sometimes beam-size k is confounded with top-k answers. Let’s not do that.
> > > >
> > > > **Question**: the comments above explain the difference between k and the tree-max-width. To be completely sure: in all the algorithms, does expanding a node/state produce *tree-max-width* states?
> > > >
> > > > If so, then I suggest saying in 3.1 that the state space is determined by the LLM policy that requires a hyper-parameter that we will call the tree-max-width.
> > > > And say in 4.1 that the tree-max-width defines the state space for all the algorithms. This is simpler to explain by moving inter-tree to a later question or the appendix. I think it’s a distraction. I understand the temptation given the interesting performance of MCTS-alpha inter-tree in Fig 3, Game 24. But that shouldn’t matter in a paper that is not trying to claim SOTA.

---

> > > > > ### Comment · Reviewer_E6Fb · 2023-11-16
> > > > > **Task parameters — a key issue**
> > > > >
> > > > > Thank you for your comments on the parameter selection. This is subtle but important.
> > > > >
> > > > > One thing is changing the parameters of the algorithm, and something else is changing the parameters of the task. Table 1 is like a description of the dataset. The amount of train and test data is given in the dataset, then why are tree max width and tree max depth there? **Please make that clear**. I suggest moving those columns to the right of the table, adding a vertical bar to separate them, and explaining in the caption that they were tuned for this paper.
> > > > >
> > > > > It’s very hard to remove the feeling that the results might be different from other values for tree-max-width/depth. In any case, the paper must include a description of the protocol for choosing the parameters in detail, including values tested.
> > > > >
> > > > > Above you said:
> > > > >
> > > > > > For GSM8K/Game 24/ProntoQA, we first experiment with a small tree-max-width and increase it to see the performance gain by increasing the search space upper bound.
> > > > >
> > > > > I don’t know what’s small and how much is the performance gain. I do know that changing the tree-max-* parameters changes the state space and that search algorithms behave differently in different search spaces.
> > > > >
> > > > > Moreover, which performance gain was observed? If the paper is accepted, the reader might wonder if the gain of the MCTS variants was more important in the selection.
> > > > >
> > > > > Above you said:
> > > > >
> > > > > > This is why in Table 3, width=50 is better than width=20 but we still choose width=20 in Table 1
> > > > >
> > > > > Table 3 can also be read as if one is satisfied with under 50% performance, the algorithm doesn’t matter. If one needs over 70% performance, the algorithm doesn’t matter. However, there are some state spaces where the proposed algorithms might offer an advantage.
> > > > >
> > > > > **Determining the width and depth of a problem is not trivial, at all**. They change the task because they change the state space. The fact that the authors considered it easy for the dataset tested in the submission. Suppose we were using these algorithms for code generation or for task-oriented dialogue. It’s plausible that the different samples require a diversity of depth, and perhaps of width. If the paper were concerned with searching within the same state-space, and results were presented with different tree-max-width/depth, then the takeaway is that if we have a way to tune those parameters in the task, then the algorithms become relevant.
> > > > >
> > > > > I think making this clear would make the paper stronger. That would also justify better the notion of comparing the same number of tokens, as they are part of the same state space.
> > > > >
> > > > > Conclusion-wise, the observations for RLHF and Game24 are more clear. For the first, the default setting was used (make sure you cite who set that value to 50). And for Game24 we get some different scenarios. For the other two tasks, I just don’t know.
> > > > >
> > > > > I’m not sure if we should accept the paper without a table similar to Table 3 but for GSM8k and PrOntoQA. Even then, we need a protocol of how the specific widths were selected. (By the way, please add to section E qualitative examples for GSM8k and PrOntoQA. According to Table 1, they should be small).
> > > > >
> > > > > The wall time in section D.9 shows that the time for Game24 and  PrOntoQA is comparable, so perhaps that’s feasible there.
> > > > >
> > > > > Questions about D.9: Is this the wall time of all the predictions or a mean per task? That’s not clear in some parts of the body of the paper when the number of tokens is discussed. Please double-check that.

---

> > > > > > ### Comment · Reviewer_E6Fb · 2023-11-16
> > > > > > **Other comments**
> > > > > >
> > > > > > At this point, I still think the submission could be better at avoiding confusion about k and path@N. path@N is about the path but it doesn’t describe the expansion of a state. By the end of section 3.2, this should be clear.
> > > > > >
> > > > > > Let me add something to my previous comment titled “Remove inter-tree for a clear state space?”: The comment in D.5 mentions only mentions CoT-SC-Tree. Perhaps it should also mention the other CoT algorithms.
> > > > > >
> > > > > > Another addition: the caption of Table 2 says “Sentence-level tasks still have variations by sampling while the token-level task is deterministic”. For sentence-level, the state space changes with the random seed and the order in which the expansions are done. For intra-tree, the sample gets fossilized once the LLM is called, so I suggest emphasizing this when referring to Table 2.
> > > > > >
> > > > > > Finally, I invite the authors to discuss the engineering challenge. Even better if that discusses the literature and/or popular libraries like Hugging Face or Triton. If this paper turns out to be influential, it might lead to optimizing LLM libraries so decoding supports the flexibility required for reducing the wall time of algorithms that in principle might be more effective. GPU memory utilization is a key issue: it might have contributed to making beam search the default choice, as beams might be easier to vectorize as sparse expansions revisit other parts of the search space. Future hardware might provide better support. Nvidia’s push for supporting a form of sparsity is motivated by the actual requirements of the algorithms.
> > > > > >
> > > > > > I'll stop my comments for a couple of days. I think another round should make this clear. Please take your time.

---

> > > > > > > ### Comment · Reviewer_E6Fb · 2023-11-17
> > > > > > > **Summary**
> > > > > > >
> > > > > > > This is a quick note to summarize my request in the long comments posted above after your last answer.I
> > > > > > >
> > > > > > > From my perspective, **the key missing pieces are**:
> > > > > > >
> > > > > > > 1. Clarify that the search space depends on the parameters max-width and max-depth, and that is *not* a parameter of the algorithm.
> > > > > > > 1. Confirm whether in intra-tree the result of the LLM policy is fossilized also for sentences. That is, for a single sample, the LLM is called only once per input, and then it’s all reused. Remark that as each algorithm expand nodes in different order, even with the same seed, the state space is different, but at least is fixed and of similar size without chance to resampling.
> > > > > > > 1. A table similar to Table 3 but for GSM8k and PrOntoQA. If one of them is missing, then that dataset should be presented as an extra result or limited experiments. Perhaps as an additional question in the result section. When presented the tasks, mention that such domain, the experiments were limited.
> > > > > > > 1. A more detailed description of the protocol for setting the max-width, including how additional values for max-width are selected for the new variations of table 3.
> > > > > > > 1. Clarifying that comparison among search algorithms is better within the same state space, and that different width might lead to different trade-off. No SOTA.
> > > > > > > 1. Make a decision about inter-tree. If there is only one algorithm doing that, then perhaps move to question in 4.2 of to appendix. If some of the baselines is also doing inter-tree, then consider a separated table, or a) organize the rows in the current tables so comparable numbers are contiguous, and b) use different line style (continuous vs dashed?) in the figures.. In any case, clarify that the number of tokens is not comparable.
> > > > > > > 1. Clarify that the proposed models are implemented as an additional head on top of the same representation, so the models cannot be used in another LLM without re-training or paying the price of decoding twice with a large model. (Btw, question: when processing a sample, are the models added to the same architecture, so the new model produces both the new tokens and then estimations? Or are those independent calls with shared weights?
> > > > > > > 1. Make sure that for all the hyper-parameters, the protocol for selection is discussed, as well as the sensibility to the parameter. For instance, in D.4 sometimes the same parameters are used for 3 task, and a different value for another one ($c_{init}$, number of simulations before setting an action). In particular, discuss the computation cost of setting those values.
> > > > > > >
> > > > > > > Extra points.
> > > > > > > * There is no need to test in other LLMs. My point was that the paper should be written as it happened: it was always the same representation, and that’s ok. Cite state of shared weights in AlphaZero and other work.
> > > > > > > * Table 3 has no variance but the appendix says that 3 seeds were used.
> > > > > > >
> > > > > > > There are other actionable requests and comments in my previous comments, please address them in a revised version too. For instance, the lack of qualitative examples for 2 of the 4 datasets.

---

> > > > > > > > ### Author Response · Authors · 2023-11-17
> > > > > > > > **Thank you for your summary**
> > > > > > > >
> > > > > > > > Dear Reviewer, thank you for your summary and your diligent and responsible reviews. We greatly appreciate your commitment and hope to see more responsible reviewers like you at ICLR.
> > > > > > > >
> > > > > > > > For your questions, we are working hard to add experiments and revise the paper to address your point. Please stay tuned for one or two days and we will come back with our updated version.

---

> > > > > > > > > ### Author Response · Authors · 2023-11-19
> > > > > > > > > **Response to Reviewer E6Fb (1/3)**
> > > > > > > > >
> > > > > > > > > We are back and we want to express our gratitude again. Thank you for your diligent and responsible reviews and we do think our paper's quality has improved a lot thanks to your suggestions.
> > > > > > > > >
> > > > > > > > > **About tree-max-width:**
> > > > > > > > >
> > > > > > > > > Thank you for making your concern clear. We finally get your point: you are worried about whether the algorithm comparisons are heavily influenced by how we set tree max-width. And you think it is possible that we draw the comparison by carefully tuning these hyperparameters.
> > > > > > > > >
> > > > > > > > > Our answer is no. We do not purposely tune this hyperparameter to make an MCTS-like algorithm the best. As we said, in fact we do not care whether MCTS-like algorithms are SOTA or not. We care about equally presenting their pros and cons. The main reason we tune it is not for algorithm comparisons but for the tradeoff between computation and performance for all tree-search algorithms.
> > > > > > > > >
> > > > > > > > > And here is our answer to your questions:
> > > > > > > > >
> > > > > > > > > **Question 1: Clarify that the search space depends on the parameters max-width and max-depth, and that is not a parameter of the algorithm.**
> > > > > > > > >
> > > > > > > > > We make further clarifications and discuss these two algorithm-agnostic parameters and how different action spaces can influence them, in the last paragraph of Section 3.1. We also address the point you mention about Table 1. We move the depth and width hyperparameters to the right and mention that they are hyperparameters for determining search space.
> > > > > > > > >
> > > > > > > > > **Question 2: Clarifying that comparison among search algorithms is better within the same state space, and that different width might lead to different trade-off. No SOTA.**
> > > > > > > > >
> > > > > > > > > Thank you for your suggestions. We have updated it in the analysis of Q2.
> > > > > > > > >
> > > > > > > > > **Question 3: And, in all the algorithms, does expanding a node/state produce tree-max-width states?**
> > > > > > > > >
> > > > > > > > > Not necessarily. As we mentioned in Section 3.1 sentence-level node subsection, for each non-terminal node, the search tree is expanded by sampling tree-max-width actions and dropping the duplicated ones. So the main reason is that we drop the duplicated one.
> > > > > > > > >
> > > > > > > > > **Question 4: A table similar to Table 3 but for GSM8k and PrOntoQA.  A more detailed description of the protocol for setting the max-width, including how additional values for max-width are selected for the new variations of table 3.**
> > > > > > > > >
> > > > > > > > > Thanks for your comments. We updated the results for GSM8K/PrOntoQA in Appendix D.6. In our submission, for the ablation of tree-max-width on Game24, we did not test it on 3 seeds, but now we’ve added the results of all 3 tasks on 3 seeds and report the average value and standard deviation in Table 7,8,9 and we modified the value of Table 3 to the mean value on Table 8.
> > > > > > > > >
> > > > > > > > > We also add more analysis about the results and as well as the protocol about how we select the width parameters. Please refer to Appendix D.6 for them.
> > > > > > > > >
> > > > > > > > > **Question 5: Table 3 has no variance but the appendix says that 3 seeds were used.**
> > > > > > > > >
> > > > > > > > > Thank you for pointing this out. This is because we conduct 3 seeds experiment for Table 2 but for Table 3 it is mainly 1 seed result. To address this, we also update the results in Table 3 (only report mean because of space limitation) and refer to Appendix D.6 for full 3 seeds results (mean an std in Table 7,8,9).
> > > > > > > > >
> > > > > > > > > **About intra-tree and inter-tree:**
> > > > > > > > >
> > > > > > > > > **Question 6: Confirm whether in intra-tree the result of the LLM policy is fossilized also for sentences. That is, for a single sample, the LLM is called only once per input, and then it’s all reused. Remark that as each algorithm expands nodes in different order, even with the same seed, the state space is different, but at least is fixed and of similar size without chance to resampling.**
> > > > > > > > >
> > > > > > > > > Yes. Once the sentence node is expanded, the corresponding sentence is fossilized and will be reused in future searches. And also you are right, since different algorithms expand differently, the actual state space is different even with the same seed.
> > > > > > > > >
> > > > > > > > > We address your point by adding a footnote 3 in Table 2’s captions and description in the second-to-last paragraph (Page 4) in Section 3.1
> > > > > > > > >
> > > > > > > > > **Question 7: Inter-tree writing problem.**
> > > > > > > > >
> > > > > > > > > Thank you for the suggestion. We move the setting of inter-tree to Appendix D.5 and hope this can make the paper easier to follow.

---

> ### Author Response · Authors · 2023-11-19
> **Response to Reviewer E6Fb (2/3)**
>
> **About the Value function:**
>
> **Question 8: Clarification that the proposed models are implemented as an additional head on top of the same representation. When processing a sample, are the models added to the same architecture, so the new model produces both the new tokens and then estimations? Or are those independent calls with shared weights?**
>
> Thank you for your suggestions. We choose to use the term “LLM-based” value in our paper to help clarify that we use an LLM based value function. We also have made more clarifications in Section 3.2.1 and the model and training detail subsection in Section 4.1. Specifically, we describe that “typically, LLM value’s decoder is adapted from original LLM policy $\pi_\theta$’s decoder, or alternatively, the LLM value $v_\theta$ and policy $\pi_\theta$ can have a shared decoder”.
>
> Let me clarify our current implementation. In our current implementation, we use a separate LLM policy and LLM value function and their decoders are not shared. This is because our critic learning needs to have training samples from the SFT policy (So our critic can be seen as an on-policy value function for the SFT sampling policy). Our experimental procedure is: We first finetune the base LLM using CoT examples in the training set, to form SFT policy. Then we sample from the SFT policy to construct the critic learning training examples. And finally we get our LLM-based value function by finetuning the base LLM (with additional value heads) using these critic training examples. Though the decoders are not shared among policy and value, they are adapted from the base model. So they may still be quite similar.
>
> And we agree with you that the shared structure might be better for computational efficiency since you do not need to call the model twice and maybe a shared decoder can get rid of all burdens brought by value function evaluation (you can reuse most computation from the policy generation). We also mention it in Appendix D.11 when discussing engineering potentials. There are several ways to achieve it. We can:
> 1. Instead of using SFT-tuned model, we can use few-shot prompted based model to generate the samples for value function training. And then we train the SFT tuned policy and value function with shared decoders.
> 2. We have some offline data that help me train the value function in advance.
> 3. We are doing the training in an online manner so the critic function can be iteratively trained, similar to the setting in AlphaZero.
>
> Currently, we are working on adding a new experiment about the shared decoder by using the same data we train the separate value and policy LLM. We want to show how such model sharing might influence the performance and also efficiency.
>
> **Question 9: There is no need to test in other LLMs. My point was that the paper should be written as it happened: it was always the same representation, and that’s ok. Cite state of shared weights in AlphaZero and other work.**
>
> Thank you for your suggestions. Please refer to our response in Q8 and we have cited the state of shared weights in AlphaZero (Figure 1.b in AlphaZero’s paper).
>
> **Other topics:**
>
> **Question 10:  Make sure that for all the hyper-parameters, the protocol for selection is discussed, as well as the sensibility to the parameter.**
>
> We add a new subsection in Appendix D.10, to fully discuss our protocol for choosing all tree-search-related hyperparameters in our paper.
>
> **Question 11: Discussion about engineering challenge.**
>
> Thank you for your suggestion. We added 5 points in Appendix D.11 (previous D.9) about the engineering challenges and how we can further increase the efficiency with more engineering efforts.

---

> > ### Author Response · Authors · 2023-11-19
> > **Response to Reviewer E6Fb (3/3)**
> >
> > **Question 12:  In appendix D.9 (now Appendix D.11), Is this the wall time of all the predictions or a mean per task? That’s not clear in some parts of the body of the paper when the number of tokens is discussed. Please double-check that.**
> >
> > Thank you for pointing this out. The previous wall time is the time for inference on all tasks. We have updated the Table with a new column to present the average time in Appendix D.11 to further clarify this.
> >
> > **Question 13: The question about k, tree-max-width and Paht@N.**
> >
> > Thank you for your suggestion, we have modified the symbol and now w refers to tree-max-width, N refers to the number of complete answer paths, and k refers to the beam size in BFS, We also added a new footnote on Page 5 to clarify the relationship between k and N in BFS.
> >
> > **Question 14: Qualitative examples for 2 of the 4 datasets.**
> >
> > We have added the rest qualitative examples for ProntoQA and GSM8K in Appendix E Table 20/21.
> >
> > **Question 15: BFS/DFS naming problem.**
> >
> > We have modified their names to BFS-V/DFS-V in our paper.
> >
> > **Question 16: Aggregation naming problem.**
> >
> > We suddenly realize your misunderstanding and want to further clarify that the aggregation process not only selects one example (only ORM-MAX selects one path). For majority-vote and ORM-vote, we need to extract each path’s final answer and conduct an answer vote (majority-vote) or weighted-vote (orm-vote and weight is determined by the orm) to determine which answer takes the largest vote. So it can be seen as one kind of combination of answers from different answer paths.
> >
> > If you still think aggregation is not the right term, what do you think of terms like answer ensemble or answer reranking?

---

> > > ### Comment · Reviewer_E6Fb · 2023-11-19
> > >
> > > Thanks. This is very thorough. I’ll check in detail later. Meanwhile, on Q8: do those fine-tuning freeze any layer or update everything? In any case, discussing that would be insightful. Look at the success of LORA and other methods that offer cheap fine-tuning.
> > >
> > > I’m looking forward to seeing results over different tree-width for the other two datasets. The reader should see them so they judge themselves, so the possible doubt about fine-tuning the treewidth disappears. It might be good to see cases if that happens, where MCTS has disadvantages.

---

> > > > ### Comment · Reviewer_E6Fb · 2023-11-19
> > > >
> > > > Aggregation: good point. That’s technically like a marginal but don’t bother formalizing that.  Just mentions that the aggregation can just select and answer or do another operation.

---

> > > > > ### Comment · Reviewer_E6Fb · 2023-11-20
> > > > >
> > > > > I went over the PDF. It’s more clear now.
> > > > >
> > > > > Regarding the new tables like table 3, I think the analysis needs more insights. For instance, it turns out that PrOntoQA is almost saturated. At that point it might not matter which algorithm works better, so it's not very informative. For the analysis, perhaps you might want to take into account the number of tokens.
> > > > >
> > > > > It wasn't clear to me in this last quick read whether the numbers of tokens are correctly account for the @10 baseline. I don't remember if that was a fair comparison or not.
> > > > >
> > > > > Just make sure everything is clear so the information is at hand for the internal discussion among the reviewers.
> > > > >
> > > > > I won't check for further changes before that. Meanwhile, thank you for the constructive dialogue.

---

> > > > > > ### Author Response · Authors · 2023-11-20
> > > > > >
> > > > > > We are a little bit confused about what you mean by saying 'whether the numbers of tokens are correctly accounted for the @10 baseline. I don't remember if that was a fair comparison or not.' Could you elaborate more on this?

---

> > > > > > > ### Comment · Reviewer_E6Fb · 2023-11-20
> > > > > > >
> > > > > > > That’s fair. My comment was confusing. I revisited Q1 in the paper: please ignore my comment about @10. That was precisely for making a more appropriate comparison.
> > > > > > >
> > > > > > > By the way, Fig 3a, bottom, says #Forward instead of #token. That should be fixed or explained in the caption. Is best-of-three? I didn’t find that anywhere else in the PDF using search.
> > > > > > >
> > > > > > > Going back to the results: Q1 in the paper is asking the question for a fixed tree-width. Q2 asks about other tree-width. I’m okay with that order as far as the conclusions of the other tree-widths are in the body of Q2. The paper looks more acceptable if Q2 confirms what was shown in detail in Q1. Otherwise, some results hidden in D.6 should have been in the body of the paper.
> > > > > > >
> > > > > > > Phrases like this should be revised to emphasize that this is not a parameter of the algorithm.
> > > > > > >
> > > > > > > > The almost doubled performance boost from 43.8 to 80.7 indicates the **importance of appropriate expansion size and tree max-width**, improving TSLLM’s performance upper bound.
> > > > > > >
> > > > > > > I’d be ok with “impact of different expansion size and tree-width”. I won’t check the whole paper for them. Please do so.
> > > > > > >
> > > > > > > ## Effective search space
> > > > > > >
> > > > > > > I mean, in general, imagine a set of 3 problems where most of the likelihood of prediction of the LLM goes to an unknown k1, k2, k3 sentences/tokens. In that case, setting the tree-width larger than ki won’t help, and reducing the under ki would be detrimental. Of course, more problems don’t have a fixed ki, and many algorithms can be bad at recovering from early mistakes (the hope is a variation of MCTS or A*, or restarts/inter-tree). (This analysis is typical of the search literature, by the way. For a recent paper about beam search, see Sofia Lemons, Carlos Linares López, Robert C. Holte, Wheeler Ruml: Beam Search: Faster and Monotonic. ICAPS 2022: 222-230)
> > > > > > >
> > > > > > > So, as you are not looking for SOTA and not exploring systematically the best tree-width, then you can only conclude about what you observed. Do not give too much importance to the chosen tree-width.
> > > > > > >
> > > > > > > ### Back to Q2 in the paper
> > > > > > >
> > > > > > > **Required**: The main conclusions in D.6 should be in Q2. Reference Q2 and Table 3 from D.6.
> > > > > > >
> > > > > > > Personally, I think the search algorithms should matter less shallow the tree, as shallow trees become more of a classification problem.
> > > > > > >
> > > > > > > Too bad PrOntoQA is not very informative in the experiments with other tree-widths. This might be the case where the effective tree-width is just low. Perhaps this would behave differently with an LLM with lower capacity, but that’s out of the scope of the current paper. I’d try that if the paper were rejected, or add it on your own if it gets accepted.
> > > > > > >
> > > > > > > That's it. Thanks again.

---

> > > > > > > > ### Author Response · Authors · 2023-11-20
> > > > > > > >
> > > > > > > > Thanks for the comment. Could you elaborate more on what phenomenons/conclusions for different tree max-width and search algorithms you hope to observe for the additional experiments with lower capacity LLM on ProntoQA? We can try that by SFT our base LLaMA with fewer training set examples but just want to make sure we are on the same page.

---

> > > > > > > > > ### Comment · Reviewer_E6Fb · 2023-11-20
> > > > > > > > >
> > > > > > > > > I meant trying, for instance, PrOntoQA with Llama2-7b and different tree-width. Perhaps despite the lower performance, there might be differences across algorithms.

---

> > > > > > > > > > ### Comment · Reviewer_E6Fb · 2023-11-20
> > > > > > > > > >
> > > > > > > > > > Beyond PrOntoQA, I meant that across different tree-width, the proposed algorithms can sometimes offer an advantage, and the disadvantages are no so significant when they appear. In any case, whatever is your conclusion in D6, it should be in Q2.

---

> > > > > > > > > > > ### Author Response · Authors · 2023-11-20
> > > > > > > > > > >
> > > > > > > > > > > Get it. We will update Q2 to include such a thing. For the PrOntoQA problem, we think it has already presented clear improvement over BFS (especially width=3, 6) compared with the rest 2 reasoning tasks: GSM8K and Game24. To demonstrate the effectiveness of MCTS-alpha/MCTS-Rollout, the RLHF experiment is the most appropriate and effective case since the search tree is much deeper and wider than the rest three tasks. Basically, you can check Table 2 and Figure 3 for the RLHF performance. Only MCTS-alpha/MCTS-Rollout can work while BFS/DFS/MCTS completely fail and are even worse than the CoT/CoT-SC baselines.
> > > > > > > > > > >
> > > > > > > > > > > We want to thank you again for this long but constructive and helpful dialogue. Meanwhile, we respectfully and kindly ask whether you can reconsider your score if our response and revision help address your concerns.

---

### Official Review · Reviewer_Zruf · 2023-11-10

**Soundness:** 3 good
**Presentation:** 4 excellent
**Contribution:** 3 good
**Rating:** 6
**Confidence:** 3

**Summary:**

The work describes a new framework called TS-LLM that enhances the reasoning abilities of large language models (LLMs) during both inference and training. Traditional approaches like Chain-of-Thought and Tree-of-Thought rely on human-designed prompts and methods like beam search or sampling, but they are limited in scope and scalability. TS-LLM overcomes these limitations by incorporating a tree-search algorithm guided by a learned value function, similar to AlphaZero. This framework is versatile, applicable to various tasks, and can work with LLMs of any size without the need for complex prompting strategies. Its effectiveness is demonstrated through empirical evaluations in different areas, including reasoning, planning, and RLHF alignment, showing that it can handle complex tree searches up to 64 steps deep.

**Strengths:**

- The experiment results are impressive.
- The framework can work on different kinds of tasks.
- The method makes sense.

**Weaknesses:**

- After ToT, using MCTS to search is not too novel. This paper uses MCTS directly.  However, I think it can still be accepted if this is the first paper that successfully uses MCTS.
- This paper needs to have more experiments to show the improvement. For example, more models and more tasks. For more tasks, maybe you can find some here “https://github.com/Ber666/llm-reasoners”.
- The paper should discuss which kinds of tasks are better for using MCTS rollout and which are better for MCTS alpha.

**Questions:**

Have you tried using both rollout and value together? If the speed of rollout and the speed of the value network are very different, you can reference the paper “Multiple Policy Value Monte Carlo Tree Search.”

---

> ### Author Response · Authors · 2023-11-13
> **Response to Reviewer Zruf**
>
> We thank you for your valuable feedback and comments.
>
> **Question 1: This paper needs to have more experiments to show the improvement. For example, more models and more tasks. For more tasks, maybe you can find some here “https://github.com/Ber666/llm-reasoners”.**
>
> Thank you for your suggestions. In fact, we do refer to “https://github.com/Ber666/llm-reasoners” for some of our experiments including GMS8K, ProntoQA and Game24. We also add a brand new RLHF task to demonstrate the general applicability of our TS-LLM framework. You can have a more detailed look at their repo and it turns out that most environments they provide do not contain any benchmark results (the table in their Experiment Results section in README).
>
> **Question 2: The paper should discuss which kinds of tasks are better for using MCTS rollout and which  are better for MCTS alpha.**
>
> Thank you for your comment. The only difference between MCTs-Rollout and MCT-alpha is that MCTS-rollout conducts the search from the beginning while MCTS-alpha conducts the search from the last search node. We think MCTS-rollout is just an offline version of MCTS-alpha since it reconducts the full search from the beginning every time. Our conclusion is, basically MCTS-rollout and MCTS-alpha will be good at the same kind of task, while MCTS-rollout can scale up token consumption to search for better generation. According to your suggestion, we have added a discussion to the MCTS-Rollout description in section 3.2.2.
>
> **Question 3: Have you tried using both rollout and value together? If the speed of rollout and the speed of the value network are very different, you can reference the paper “Multiple Policy Value Monte Carlo Tree Search.”**
>
> Thank you for your comment and we will refer to this paper. Currently, we use two separate networks to conduct rollout and value generation respectively and the speed of rollout and the speed of value estimation are similar. But note that we can also utilize a shared LLM decoder for both policy and value network so the rollout and value estimation can be conducted simultaneously and be more efficient.

---

> > ### Comment · Reviewer_Zruf · 2023-11-16
> > **Comments**
> >
> > Q1:
> > I know that you have included some of the task. I was just saying that it will be better to have more. Also you didn't replay the using different models part.
> >
> > Q2:
> > I am confused. When we talk about rollout in MCTS, it normally means that after we select a leaf node, we conduct a rollout (simulation) from the leaf node with a rollout policy instead of evaluated the leaf node with a value function. Are we on the same page?
> >
> > Q3:
> > understood.  It will still be cool if you can show that the combining version can be best at all tasks.

---

> > > ### Author Response · Authors · 2023-11-16
> > > **Response to Reviewer Zruf**
> > >
> > > Q1: Thank you for your suggestions. We will try to add new tasks. For the model, in our experiment, we do try two different models: LLaMA2-7b and GPT-2. Because of the limitation of our computing resources, currently, we cannot try larger model. But we can still do it on a model under 7B size. Do you have any suggestions on which model we should try?
> > >
> > > Q2: There are three search algorithm variants: MCTS/MCTS-rollout/MCTS-alpha. What you are referring to (conduct a rollout (simulation) from the leaf node with a rollout policy instead of evaluated the leaf node with a value function), is the MCTS variant. As we mentioned in Section 3.2.2, MCTS-Rollout is basically similar to MCTS-alpha (which uses value function instead of rollout-based MC simulation). The only difference is that MCTS-Rollout always starts at the beginning, so it will search for the whole rollout /Path instead of conducting online search (which is MCTS-alpha in the AlphaZero paper). That is why we also think MCTS-Rollout can be seen as an offline version of MCTS-alpha since it needs to reconduct the search from the very beginning.
> > >
> > > We think the confusion mainly comes from the way we name it. Do you think it is more easy to understand if we call it offline MCTS-alpha? or do you have any other suggestions?
> > >
> > > Q3: Thank you for your comment. We will try to add the combined version.

---

> > > ### Author Response · Authors · 2023-11-20
> > > **New experiment update on Q3**
> > >
> > > Dear Reviewer Zruf,
> > >
> > > We have added a new section in Appendix D.12 to address your Q3 by new experiments where the policy and value network share the LLM decoder. We systematically analyze its performance and efficiency in Table 19 and 20. And thanks for your suggestions! The computational efficiency increase is quite large! (20x in the token-level case while 9x in the sentence-level case for value calculations). We will definitely conduct more thorough experiments over the shared setting in our future work.

---

> > > > ### Author Response · Authors · 2023-11-22
> > > >
> > > > Dear Reviewer Zruf, we hope our answers and results above have addressed your concerns. If so, we would greatly appreciate it if you could reconsider your score. Please let us know if there are any other questions.

---

### Author Response · Authors · 2023-11-13
**Global Response (Updated on November 23)**

Dear reviewers,

We thank you very much for your constructive discussions and precious suggestions in helping us improve the paper.

We thank reviewer **Zruf**’s positive feedback and suggestions to help us increase our algorithm’s efficiency (details explained in point 4 below).
We are also very grateful to reviewer **xk9Q** and **1bFk** for their constructive suggestions to help us improve the clarity of our transcripts and their affirmation of our efforts on the extensive experiments and writing.

Specifically, we want to acknowledge reviewer **E6Fb** sincerely, not only for his/her patience and careful suggestions that make our transcripts much more organized and comprehensive but also for what we have gained and agreed upon about all questions and concerns during these days of heated discussions.

In this letter, we aim to summarize the consensus reached during our ten days of fruitful exchanges, outlining the key revisions made in response to your collective feedback. We have prepared a revision of our transcripts in accordance with these comments. In particular, we have made the following changes (all modified contents in our main paper and appendix are highlighted in red):
- In our methodology section, we’ve fixed the writing problems about algorithms’ names, and terminology and rearranged our writings. We’ve made modifications to better clarify different algorithms and implementation details by adding Appendix C.2 and D.5 to illustrate them in detail.
- In our experiment section, we clarify more about the experiment setups, our benchmark algorithms, and baselines in section 4.1. We clarify more about the Path@1 result mentioned in Q1’s answer and add the Path@N result as complementary to make the analysis more reasonable.
- We totally appreciate reviewer **E6Fb**’s suggestions about the computational efficiency issues. Therefore, we’ve added the wall-time result for each search algorithm in Appendix D.11 and analysis for computational efficiency, engineering challenges, and the improvements we are trying to conduct in Appendix D.11.
- For suggestions about LLM policy and value function with the shared decoder of the reviewer **Zruf** and **E6Fb**, we also conducted experiments in Appendix D.12, discussing the pros and cons of updating both policy and value loss together, as well as demonstrating the significant improvement on the computational efficiency of the shared decoder architecture.
- For reviewer **E6Fb**’s concern about the sensitivity of the hyperparameters of tree search algorithms in our framework, we’ve conducted comprehensive ablations about different tree-max-widths’ effects on 3 tasks in Appendix D.6. And we’ve added a discussion about how we select all the hyperparameters in detail in Appendix D.10.


We would like to address a commonly raised concern regarding the perceived inadequacy of MCTS-alpha/MCTS-rollout as presented in Table 2:
- **We want to clarify that TS-LLM’s DFS-V/BFS-V variants are different from those in Tree of Thought (ToT). They are TS-LLM’s variants instead of ToT baselines.** The main difference is that we use a learned value function instead of prompting advanced LLMs for self-evaluation, which is one of our important contributions. We have shown in Table 4 the result of BFS-V and MCTS-alpha on Game24 by prompting LLaMA2-7B as value function and ORM. It can be regarded as an instance of ToT over small LLMs. Our conclusion is that prompting a small LLM can not offer accurate value estimation and the performance of tree-search is even worse than the simplest CoT baseline.
- Table 2 represents path@1 results. A much more comprehensive result can be seen in our Q4 experiment. We additionally conduct experiments on search aggregation shown in Figure 3, where you can get Path@N results and scale up your token consumption by running multiple times of search in MCTS/MCTS-alpha/MCTS-rollout/DFS-V or setting larger beam size in BFS-V. Specifically, the bottom line of Figure 3 clearly demonstrates the performance of each tree-search algorithm when given the same token consumption. Basically, MCTS-alpha and MCTS-rollout are still dominant in most cases.
- Lastly, we would like to clarify that our paper does not claim MCTS-alpha/rollout as the new state-of-the-art for all evaluation tasks. Our aim is to present a balanced view of various search algorithms, acknowledging their strengths and limitations in different contexts. Our experiments are not designed to show new SOTAs or the most efficient tree search algorithm. **We treat different search algorithms as TS-LLM‘s variants and equally present their pros and cons. We believe every algorithm has its limitations and no one can claim superiority in all scenarios.**

Sincerely,

The Authors of Paper 4470.

---

### Meta-Review · Area_Chair_XeW9 · 2023-12-05

**Metareview:**

This paper shows that it is possible to train smaller LLMs to perform tree search, in the spirit of tree of thought and reasoning as planning. The architecture uses MCTS with a learned value network, and considers simple datasets popularly used to benchmark LLM inference strategies.

A strength of the work is that it shows that tuning smaller models as value functions can be competitive with GPT-4 powered tree searches (though as the reviewers point out, it has not actually set a new state of the art on these benchmarks).

A weakness of the work is that it is not very novel, in the broader context of LLM-guided tree search. The authors could address this weakness by having a new empirical result that they can accomplish but prior works can't.

**Justification For Why Not Higher Score:**

Given that this is a relatively straightforward "gluing together" of existing components in order to build a tree search algorithm for LLM inference, and given the lack of surprising new empirical advances, it is hard to recommend publishing this paper at ICLR given the current state of the work.

**Justification For Why Not Lower Score:**

n/a

---

### Decision · Program_Chairs · 2024-01-16

Reject